# The Social Backgrounds, Dharma Lineages, and Achievements of Women Chan Masters, 1572–1722

**Yuh-Neu Chen**

Department of History, National Cheng Kung University, No. 1, University Road, Tainan City 701, Taiwan; z8508054@email.ncku.edu.tw

**Abstract:** This article continues the investigation of my previous paper 'Wan Ming Qing chu dongnan yanhai gangkou Fosi de biqiuni shenying' 晚明清初東南沿海港口佛寺的比丘尼身影 ('The Historic Image of the Bhikkhuni Who Lived at Buddhist Monasteries of Seaport Cities of Southeast China during the Late Ming and Early Qing Period'). While discussing the space of port city Buddhist monasteries and their urban environment, and how they aided or hindered the bhikkhuni monastic community, or individual bhikkhuni in their practice, life, and personal achievements, I also realized that a bhikkhuni's family background and her connections with local gentry and eminent persons indeed contributed to the rise of her prestige. The publication and distribution of the recorded sayings of several women Chan monastics residing in the Buddhist monasteries of port cities during the late Ming and early Qing periods can be regarded as a commendable breakthrough in the Buddhist history of this period. To further clarify the resource structure that helped support these Chan bhikkhuni, including the multifaceted interplay between their blood ties, dharma connections, and regional connections, I aim here to particularly examine the backgrounds, dharma lineages, activity regions, and ultimate achievements of these women Chan monastics. I will consider how it is that these Chan bhikkhuni were able to use such resources to achieve the distinction of having their deeds recorded, while so many more of their fellow bhikkhuni have been forgotten. While in real life, there were a rather large number of bhikkhuni, only a few have been able to have their names passed down to posterity. Women have always faced greater challenges than men in having their voices known and receiving social recognition, and even for men, this has never been easy without a relevant support system—one's own personal cultivation, the prestige of one's family, the power of one's dharma lineage, the strength of one's personal connection, the disparity in the resources that one has access to in their urban or rural society, etc.; all these various factors affect one's achievements and performance. Especially in Chan Buddhist texts, where women are greatly underrepresented, I aim to explore in this paper how a women Chan monastic could be included in the Chan historical record, have her recorded sayings published, and have a place in Buddhism or the dharma lineages of Chan.

**Keywords:** The Ming and Qing period; *nü chanshi* 女禪師 (women Chan masters); bhikkhuni; *famai* 法脈 (dharma lineages); the Chan school



## 1. Introduction

The relevant records of bhikkhuni from the Ming–Qing period who were clearly listed as being part of a certain Chan lineage may not reflect the actual situation. There were likely many more remarkable bhikkhuni of that time than recorded, yet biographies and other related textual records for bhikkhuni women are much less than those of bhikkhu men. Where such records do exist, they are scattered in various kinds of documents, such as anthologies, miscellanies, jottings, and poetry. Many of these records are also just a few words in length, making it difficult to collect and organize them. Even where there are clear records, it is often the case of where a woman would shave her own head, ordain herself, and have her own house be her convent. Compared to those of men, the records

of women who ventured to seek out a teacher of Chan to receive tonsure, the full precepts, and dharma transmission are quite limited and often unclear.

Even so, in this paper, I would like to take the materials I have previously collected and consider how some Chan bhikkhuni were able to distinguish themselves from amongst numerous other members of the Buddhist sangha, especially in terms of their dharma lineages, the locations where they were active, and their lifetime accomplishments. How were they able to have their deeds recorded and become known to posterity? This is an extension of my previous paper 'Wanming Qingchu Dongnan yanhai gangkou Fosi de biqiuni shenying' 晚明清初東南沿海港口佛寺的比丘尼身影 ('The Historic Image of the Bhikkhuni Who Lived at Buddhist Monasteries of Seaport Cities of Southeast China during the Late Ming and Early Qing Period'), and here, I will go a step further by expanding my scope of analysis to dharma lineages in my hope that this problem can receive greater attention. A common sight in Ming–Qing historical records are accounts such as, 'The bhikkhuni of the world are most abundant in Zhejiang province, there are no fewer than several hundred thousand in only the three prefectures of Hangzhou, Jiaxing, and Huzhou' (天下尼僧惟浙中最盛, 即杭嘉湖三府已不下數十萬人) (Lan 1975, 1:37) and 'There are many bhikkhuni monasteries in the capital' 京師多尼寺 (Lu 1993, 6:1682). Although there were actually many women who became bhikkhuni, those who could leave their names to posterity are few and far between. Women have always faced greater challenges than men in having a voice and receiving recognition, and even for men, this has never been easy without a relevant support system—one's own personal cultivation, the prestige of one's family, the power of one's dharma lineage, the strength of one's personal connection, the disparity in the resources that one has access to in their urban or rural society, etc.; all these various factors affect one's achievements and performance. Especially in the texts of Chan Buddhism, where women are greatly underrepresented, how could a Chan bhikkhuni come to be recorded in Chan historical records, have her recorded sayings published, and have herself a seat in the Buddhism or the dharma lineage of Chan? This is a question that merits further exploration.[1]

## 2. Chan Bhikkhuni in the Dharma Lineages of the Linji and Caodong Denominations

In recent years, I have assembled a collection of twelve different catalogs of the Jiaxing Buddhist canon from all over the world, as summarized in Table 1:

**Table 1.** Twelve catalogs of the Jiaxing canon.

| Compiler | Catalog | Place of Publication | Publication Date |
|---|---|---|---|
| Bairen ji 梅林寺 | *Kōnan zan Bairinji shōzō Kakō daizōkyō mokuroku* 江南山梅林寺所藏嘉興大藏経目録 (The catalog of the Jiaxing canon as preserved in Meilin temple of the Jiangnan region) | Kumeru: Bairen ji | 1998 |
| Xinwenfeng chuban gongsi bianjibu 新文豐出版公司編輯部 | *Songban Qisha dazang jing· Mingban Jiaxing dazang jing· fence mulu fenlei mulu zong suoyin* 宋版磧砂大藏經．明版嘉興大藏經–分冊目錄．分類目錄．總索引 (The Song-edition Qisha canon and the Ming-edition Jiaxing canon: the catalog of each volume, the catalog of each category, and the general index) | Taibei: Xinwenfeng chuban | 1988 |
| Guojia tushuguan tecangzu 國家圖書館特藏組 | *Guojia tushuguan shanben zhi chugao·zibu* 國家圖書館善本志初稿·子部 (A draft record of rare edition books preserved in National Central Library) | Taibei: Guojia tushuguan | 2000 |

**Table 1.** *Cont.*

| Compiler | Catalog | Place of Publication | Publication Date |
|---|---|---|---|
| Anonymous | *Jiaxing zang mulu yijuan* 嘉興藏目錄一卷 (The catalog of the Jiaxing canon, one fascicle) | Taibei: Chengwen chuban (photoprint of the edition published by Beijing Kejing Chu in 1920) | 1978 |
| Zhejiang daxue tushuguan 浙江大學圖書館 | *Jiaxing zang mulu* 嘉興藏目錄 (The catalog of the Jiaxing canon ) (Excel sheet) | Hangzhou: Zhejiang daxue chubanshe | n.d. |
| Yokote Yutaka 横手裕, etc. | *Tōkyō daigaku sōgō toshokan shozō Kakō daizōkyō mokuroku* 東京大学総合図書館所蔵嘉興大蔵経目録 (The catalog of the Jiaxing canon as preserved in the General Library of Tokyo University) | Tokyo: Tōkyō daigaku dai gaku in jinbun shakai kei kenkyūka | 2010 |
| Yokote Yutaka 横手裕, etc. | *Seikadō bunko shozō Zuishō ji kyūzō* 靜嘉堂文庫所蔵瑞聖寺舊蔵 (The collection of Ruiying temple as preserved in Seikadō Bunko) | *Investigation Report of Tōkyō daigaku sōgō toshokan shozō n Banrenki ba daizōkyō (Kakō zō)* 東京大学総合図書館所蔵萬曆版大蔵経(嘉興蔵)(call no.: 040-4514, Tokyo: Tōkyō daigaku daigakuin jinbun shakai kei kenkyūka): I·'Mokuroku hen' 目録篇, pp. 473–88. | 2010 |
| Xiyuan si tushuguan 西園寺圖書館 | *Xiyuan si Jiaxing zang* 西園寺嘉興藏 (The catalog of the Jiaxing canon, as preserved in Xiyuan temple) | Suzhou: Xiyuansi cangjing lou huicun 西園寺藏經樓惠存 (thus is written on the hand-copied catalog provided by the Xiyuan si monastery) | 1981 |
| Hanazono daigaku kokusai zengaku kenkyū sho 花園大学国際禅学研究所 | *Makomosan Manjū ji Shozō Kakō zō (Kakō daizōkyō) mokuroku* 蔣山万寿寺所藏嘉興藏 (嘉興大藏經)目錄 (The catalog of the Jiaxing canon, as preserved in the Makomosan Manjū monastery) | Kyoto: Kawakita insatsu, 2019 (collection of the Makomosan Manjū monastery, Japan) | 2019 |
| Shoudu tushuguan 首都圖書館 | *Shoudu tushuguan guji shanben shumu* 首都圖書館古籍善本書目 (The catalog of rare-edition books preserved in Capital Library of China) | Beijing: Guojia tushuguan chubanshe | 2011 |
| Jingshan zang bianweihui 徑山藏編委會 | *Jingshan zang* 徑山藏 (The Jingshan canon) | Beijing: Guojia tushuguan chubanshe | 2016 |
| Anonymous | *Jiaxing zang mulu* 嘉興藏目錄 (The catalog of the Jiaxing canon) | (Collection of the Thập Tháp từ 十塔寺 temple, Thị Xã An Nhơn, Vietnam) *LIÊUQUAN* 了觀, Issue: 23. Shunhua: Shunhua chubanshe 順化出版社. | May 2021 |

Here, I will consider the writings of men and women Buddhist monastics of the late Ming and early Qing—that is, from the Wanli (r. 1573–1620) through the Kangxi (r. 1661–1722) reigns—including Chan discourse records, individual commentaries, and other compositions. I will use these writings primarily to research the lives of these men and women monastics and confirm their lineages of dharma transmission. I will use as supplementary sources the *Wudeng quanshu* 五燈全書 (*Complete Compendium of the Five Lamp Transmissions of Chan Genealogy*), the *Xu biqiuni zhuan* 續比丘尼傳 (*Continued Biographies of Bhikkhuni*),[2] the *Renming guifan jiansuo ku* 人名規範檢索庫 (Buddhist Studies Person

Authority Databases), the China Biographical Database (CBDB), and relevant journal articles. I will then organize these men and women according to the relationships between their dharma lineages and by their different affiliations—such as to the Linji or Caodong denominations—and I will integrate these data into figures. The organizational focus of these figures is the Chan monastics whose works were included in the Jiaxing canon as a genealogy of dharma transmission inheritors, including their disciples of various dharma lineages who assisted in compiling discourse records or had their own writings included in the canon, but it is inevitably not exhaustive. In the following figures, Chan monastics who were men will be indicated by a black frame, those who were women will be indicated by a black frame with a yellow note of their dharma name, and those whose writings were included in the Jiaxing canon will be indicated with a red frame (including women).

Overall, a comparative view of these data allows us to see that the Linji denomination was unmatched in its dominance, and although not as many women Chan monastics were listed in the genealogies of dharma lineages as men, most of these women were affiliated with the Linji denomination. The writings of the Ming–Qing period Chan monastics that were included in the canon were also mostly of the Linji denomination, as can be seen more precisely in the figures.

The overall view of the Linji dharma lineage in the late Ming and early Qing periods can be summarized as follows:

> Tracing back until Yuanwu Keqin 圜悟克勤 (1063–1135), through Yuansou Xing-duan 元叟行端 (1255–1341), this lineage expanded in the late Ming and early Qing periods to Yueming Lianchi 月明聯池 (1574–1639) and to his dharma heir Chuiwan Guangzhen 吹萬廣真 (1582–1639). The members of this lineage resided primarily in the Sichuan–Huguang area. Guangzhen's disciples were Tiebi Huili 鐵壁慧麗 (1586–1650) and Tiebi Huiji 鐵壁慧機 (1603–1668). After Huiji, this lineage branched out to numerous disciples. However, there are no records of women disciples in this dharma lineage, as seen in Figure 1.

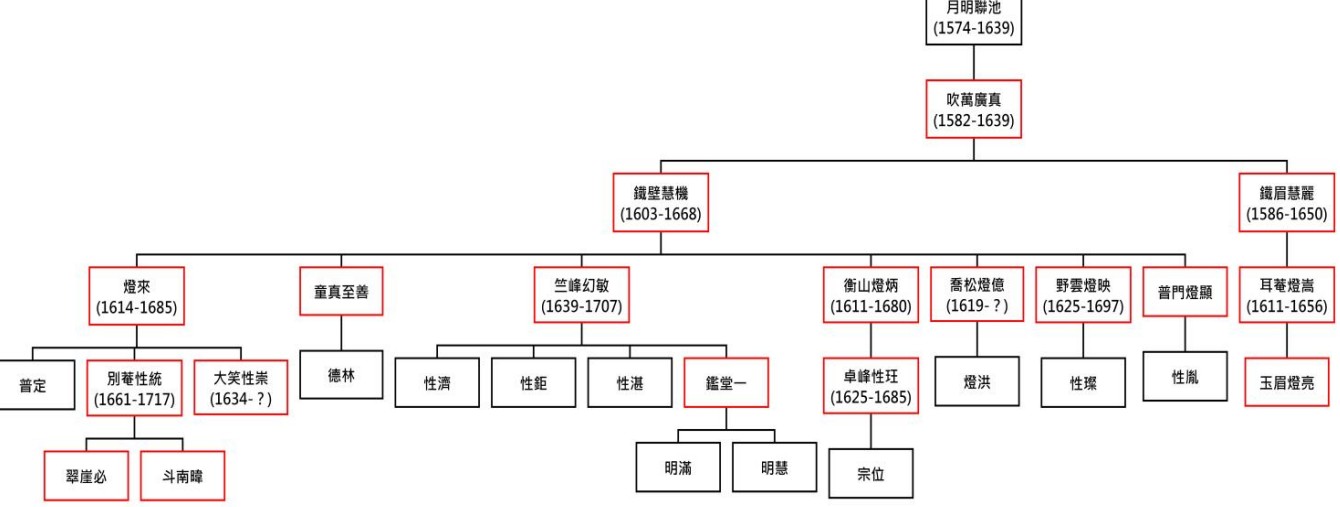

**Figure 1.** The dharma lineage of Chuiwan Guangzhen of the Linji denomination.

This lineage also traced its dharma transmission to Yuanwu Keqin but belonged to the separate branch of Hemi An 何密庵, who was a disciple of Wuzhun Shifan 無準師範 (1178–1249), and Hemi An's disciple Fulin Zhidu 福林智度 (1304–1370), whose disciple Bu'er Zhenji 不二真際 primarily propagated the dharma in the Jiangnan cities of Hangzhou and the Huguang region. There are no women disciples to be seen in this dharma lineage, as can be seen in Figure 2.

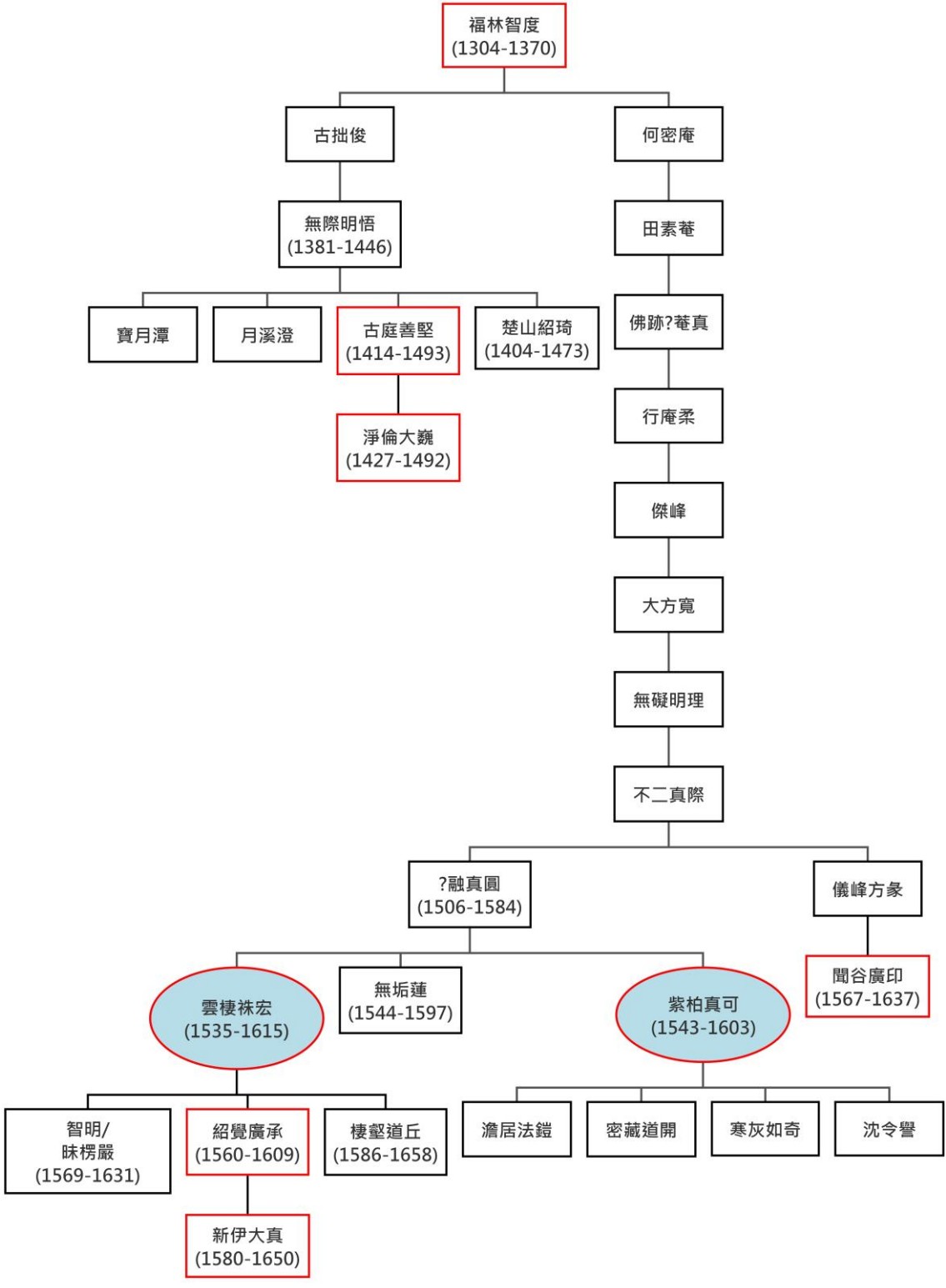

**Figure 2.** The dharma lineage of Bu'er Zhenji of the Linji denomination.

As for the other branch of Fulin Zhidu's dharma lineage, that of Guzhuo Jun 古拙俊, this lineage was transmitted to Nanming Huiguang 南明慧廣 (1576–1620) and then to Yuanhu Miaoyong 鴛湖妙用 (1587–1642). Disciples in this lineage included Jie'an Wujin 介菴悟進 (1612–1673) and Yichu Wuyuan 一初悟元 (1615–1678). This branch propagated the dharma mostly in the Huguang and Zhejiang areas but reached as far as Fujian. There

were many disciples of Jie'an Wujin, and most of them resided in the areas of Huguang, Jiangsu–Zhejiang, and Jiangxi. The women disciples who received this dharma transmission included Mingxin Foyin 明心佛音. There also was a bhikkhuni disciple of Shanduo Zhenzai 山鐸真在 (1621–1672) named Shengdi Zhuo 聖地拙. Bao'ru Yu 寶如玉, who was a disciple of Suhong Zhenli 素弘真理, had a bhikkhuni disciple named Yun Guzong 蘊古宗, but she had no writings incorporated into the canon. See Figure 3.

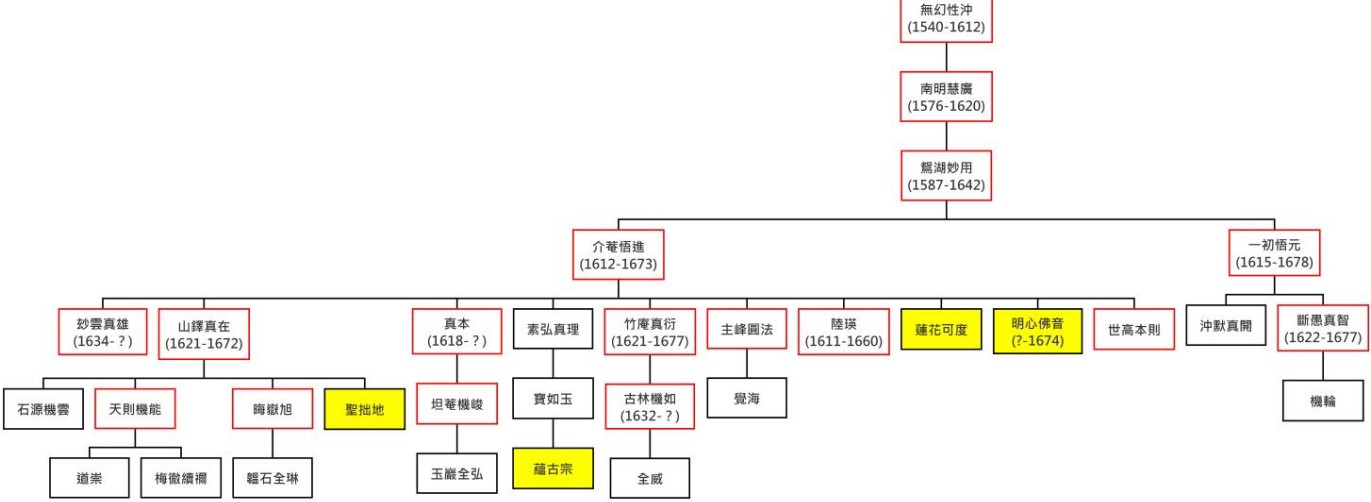

**Figure 3.** The dharma lineage of Yuanhu Miaoyong of the Linji denomination.

This dharma lineage was also traced back to Yuanwu Keqin and in the Ming through Xiaoyan Debao 笑巖德寶 (1513–1581) and later Huanyou Zhengchuan 幻有正傳 (1549–1614). Three main branches of this lineage were those of Miyun Yuanwu 密雲圓悟 (1566–1642), Tianyin Yuanxiu 天隱圓脩 (1575–1635), and Xueqiao Yuanxin 雪嶠圓信 (1571–1647). Miyun Yuanwu's lineage was the most influential of these. For details, refer to Figure 4. Tianyin Yuanxiu and Xueqiao Yuanxin primarily propagated the dharma in the areas of Jiangsu–Zhejiang, Huguang, and Jiangxi, and there are no records of there being women disciples in these two dharma lineages.

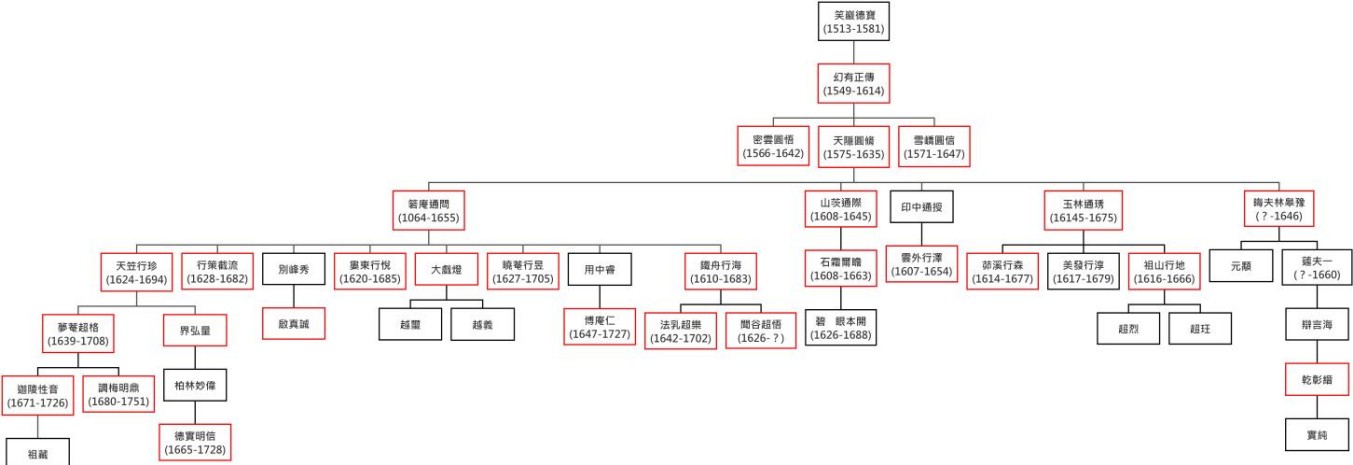

**Figure 4.** The three dharma lineages of Yuanwu, Yuanxiu, and Yuanxin of the Linji denomination.

However, Miyun Yuanwu's dharma linage flourished, as can be seen in Figure 5a ('Miyun Yuanwu's Dharma Lineage (1)') through Figure 6g ('Miyun Yuanwu's Dharma Lineage (7)'). Among Miyun's dharma heirs were Shanweng Daomin 山翁道忞 (1596–1674), Muyun Tongmen 牧雲通門 (1599–1671), Feiyin Tongrong 費隱通容

(1593–1661), Tongxuan Tongqi 通玄通奇 (1596–1652), Fushi Tongxian 浮石通賢 (1593–1667), Guxue Zhenzhe 古雪真喆 (b. 1614), Hanyue Fazang 漢月法藏 (1573–1635), Wuwei Ruxue 無為如學 (1585–1633), Poshan Haiming 破山海明 (1597–1666), Poshi Wuzhuo 破石悟卓 (1609–1654), Chaozong Tongren 朝宗通忍 (1604–1648), Longchi Tong-wei 龍池通微 (1594–1657), Shiqi Tongyun 石奇通雲 (1594–1663), Zhaojue Zhangxue昭覺丈雪 (1610–1696; also called Tongzui 通醉), Shiche Tongcheng 石車通乘 (1598–1638), and Yingning Zhijing 攖寧智靜. This dharma lineage is characterized by its continuous out-pouring of talent and also by its many Chan bhikkhuni members, especially in the dharma lineage of Shiche Tongcheng. Second to this was the dharma lineage of Shiqu Tongyun, who transmitted the dharma to his bhikkhuni disciple Weiji Xingzhi 惟極行致 (d. 1672). Xingzhi in turn transmitted the dharma to her bhikkhuni disciple Jingnuo Yue 靜諾越. There was also Fazhu Chang 法柱長, a disciple of Tongxuan, who had a bhikkhuni dis-ciple named Tongli Jing 通禮敬 (d. 1690), and Longchi Tongwei 龍池通微 (1594–1657) had a bhikkhuni dharma heir named Jizong Xingche 季總行徹 (b. 1606). See Figure 5a. These dharma heirs propagated the dharma primarily in the Zhejiang–Jiangsu region, but they also extended their teaching to other areas, such as Fujiang, Jiangxi, Huguang, Shaanxi, Shandong, Guangdong, and North Zhili.

The dharma lineage of Miyun Yuanwu's disciple Poshan Haiming also flourished, and this lineage mostly propagated the dharma in Sichuan, Guizhou, Yunnan, and Huguang, with Sichuan being the region where this lineage was especially developed. Al-though successors to this dharma lineage were abundant, like the dharma lineage of Chi-wan Guangzhen, there are not any records of it having women disciples. See Figure 5b.

There were comparatively more bhikkhuni disciples in Hanyue Fazang's dharma lin-eage. For example, Poushi Hongbi 剖石弘璧 (1599–1670) had the bhikkhuni disciple Fayu Ying 法雨瀛, and Tuiweng Hongchu 退翁弘儲 had three bhikkhuni disciples named Ren-feng Jiyin 仁風濟印, Baochi Xuanzong 寶持玄總, and Zufu Xuankui 祖符玄揆 of Lingrui 靈瑞.

Tuiweng Hongchu also had a disciple named Bo'an Zhengzhi 檗庵正志 (Xiong Kaiyuan 熊開元; 1600–1676), who had a bhikkhuni disciple named Daoyu 道遇. Also, Jude Hongli 具德弘禮 (1600–1667), who was also of the Hanyue Fazang dharma lineage, had a dharma heir named Jubo Jiheng 巨渤濟恆, who had two bhikkhuni disciples named Lingxi Rong 靈璽融 and Huizhao Lian 慧照蓮 (Gudi Lian 古滌蓮). Yimo Hongcheng 一默弘成 (1575–1641) had a disciple named Huotang Zhengyan 豁堂正喦 (1597–1670), who had a bhikkhuni disciple named Xiang'an Hui 象庵慧. The dharma transmission of each of these lineages is as shown in Figure 5c. This lineage primarily propagated the dharma in the Jiangsu–Zhejiang area but also extended to Jiangxi and Huguang.

In addition, Fushi Tongxian 浮石通賢 (1593–1667) had a bhikkhuni dharma heir named Gao Yuanqing 高源清, and Tongxian's dharma heir Fayin Xingzhi 法音行　had a bhikkhuni dharma heir named Fuhui Ji 桴海濟. Fayin Xingzhi's dharma brother Quan-shi Wo 拳石沃 had a bhikkhuni disciple named Huikong 慧空. This dharma lineage mainly spread in the areas of Guizhou, Yunnan, and Huguang. Tongxuan Tongqi 通玄通奇 (1596–1652) had a disciple named Tianmu Chaozhi 天目超智 (1626–1685), who had a bhikkhuni dharma heir named Zhaoqing Guang 照清光. The members of Tongxuan's dharma lineage resided primarily in the Jiangsu–Zhejiang area but also extended to Jiangxi and Fujian. The arrangement of Fushi Tongxian's lineage can be seen in Figure 5d.

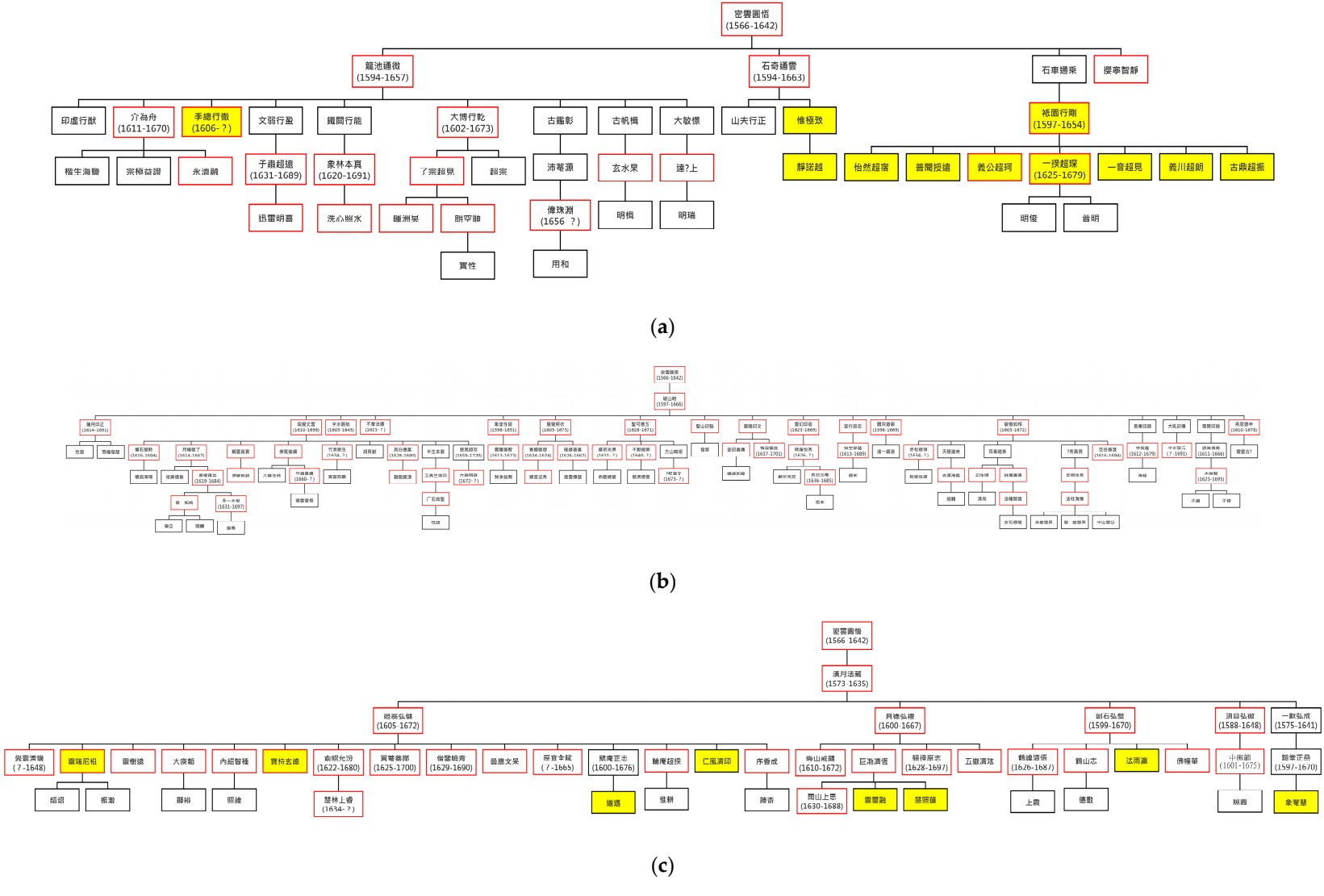

**Figure 5.** *Cont.*

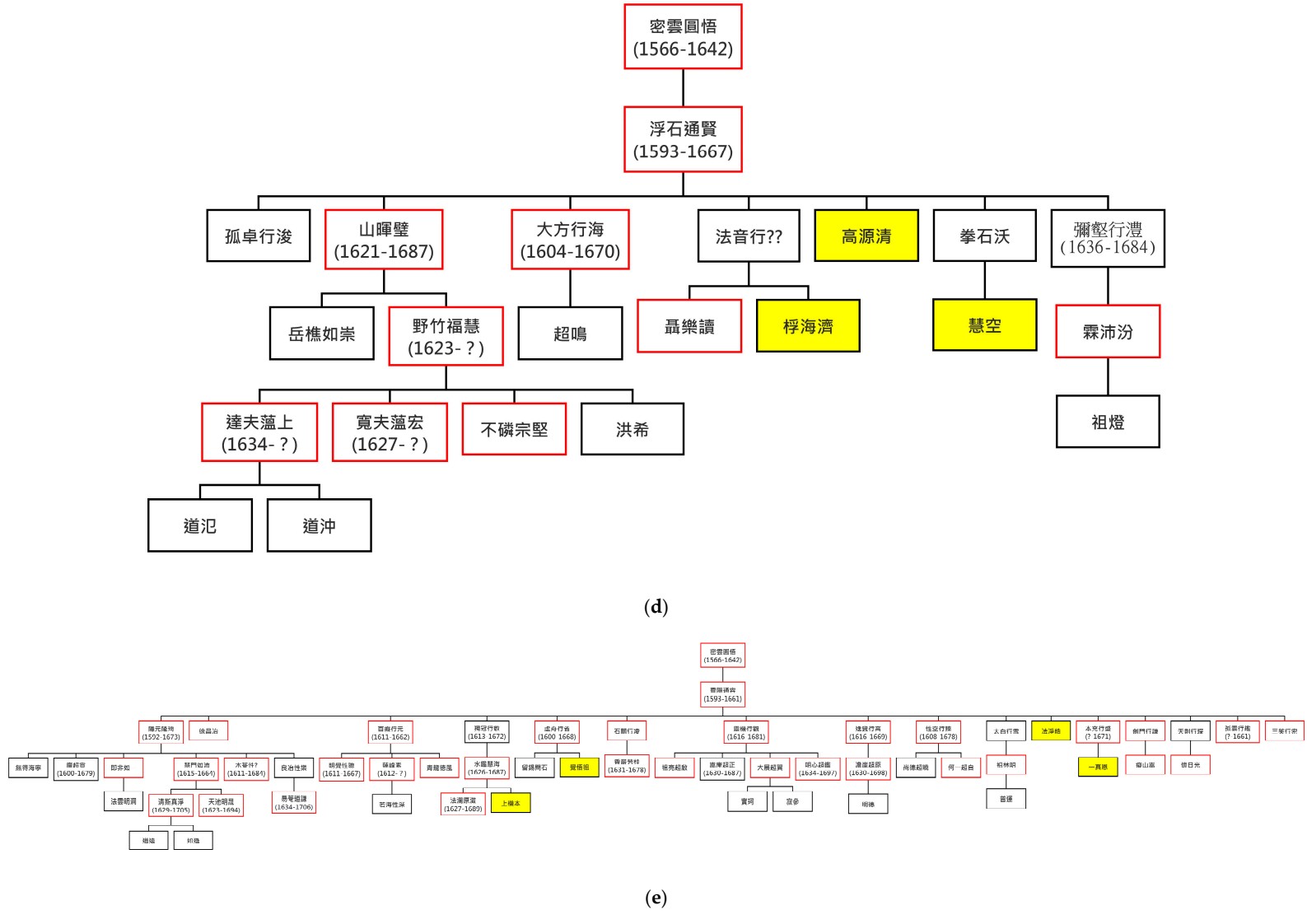

(**d**)

(**e**)

**Figure 5.** *Cont.*

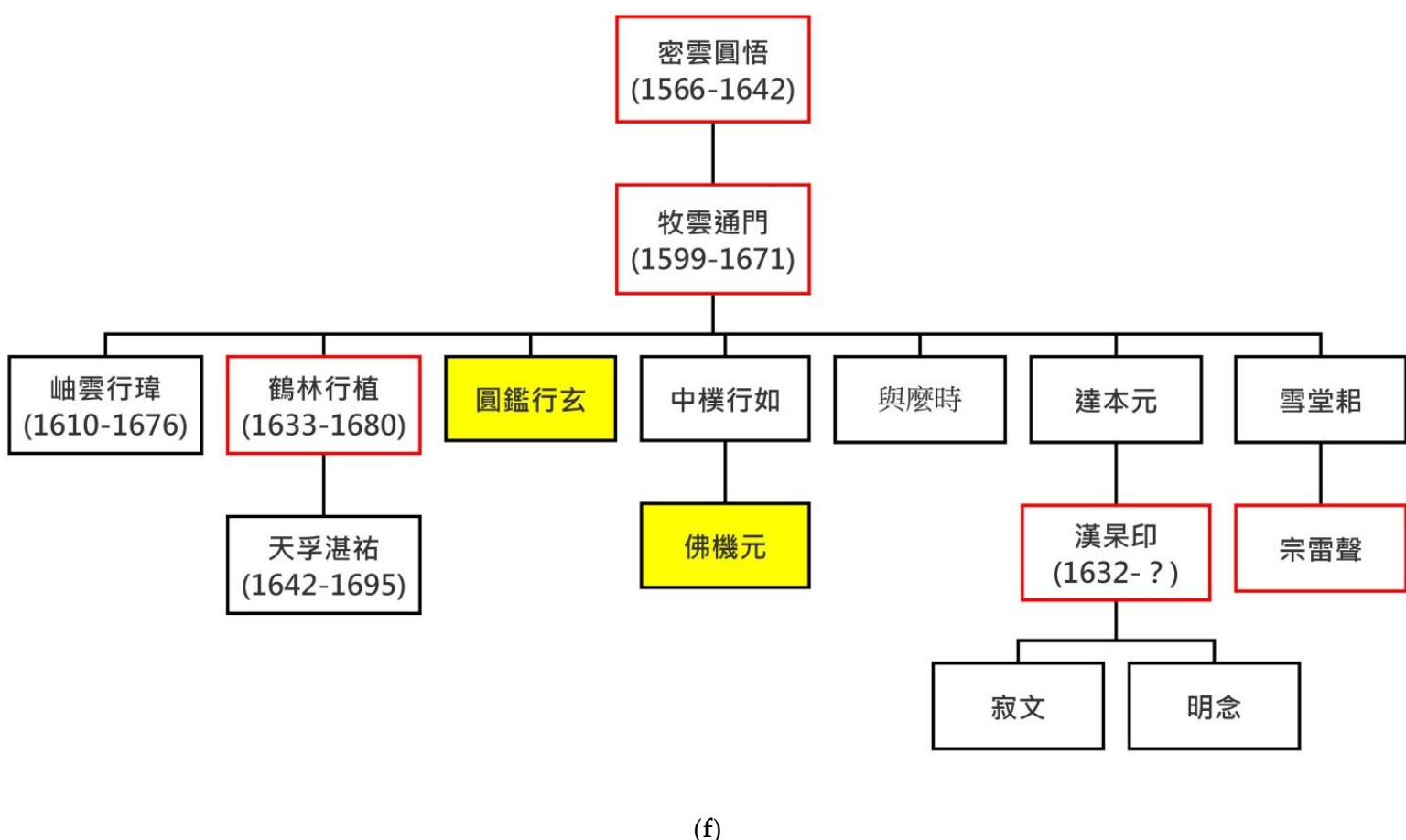

(**f**)

**Figure 5.** *Cont*.

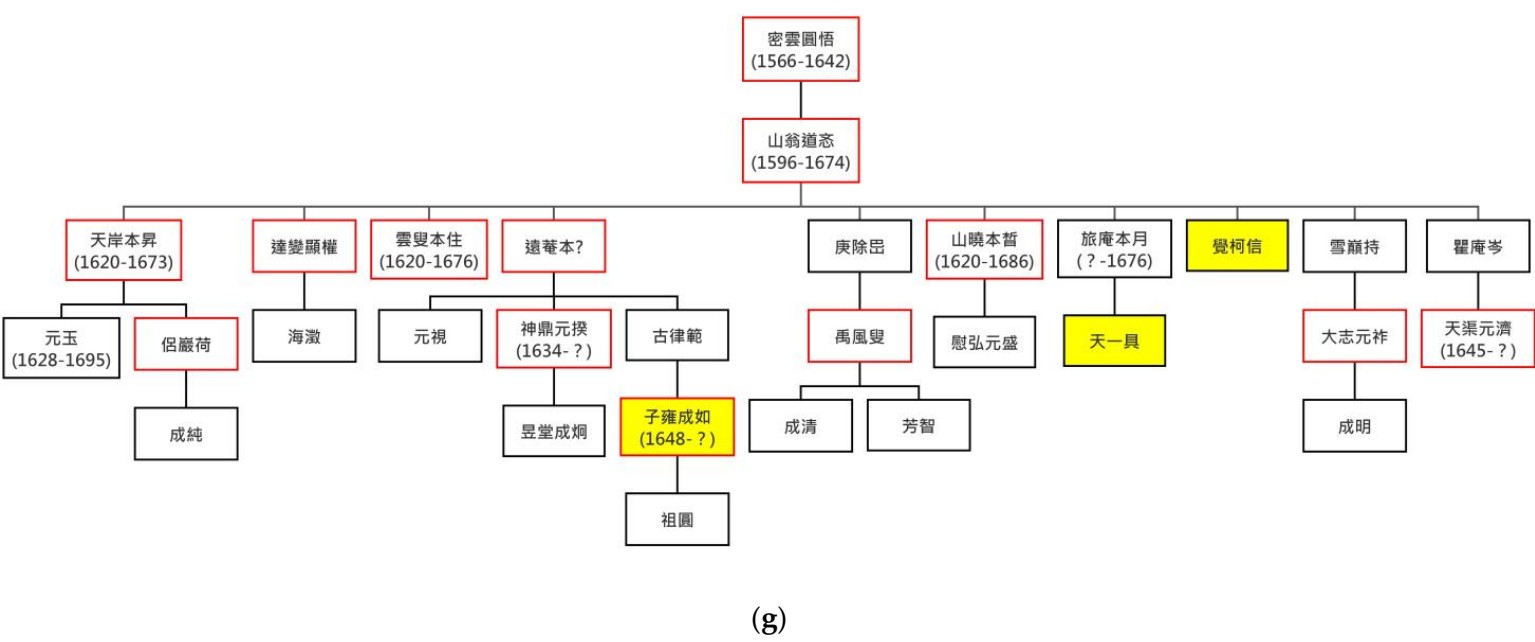

(**g**)

**Figure 5.** (**a**) Miyun Yuanwu's Dharma Lineage (1). (**b**) Miyun Yuanwu's Dharma Lineage (2). (**c**) Miyun Yuanwu's Dharma Lineage (3). (**d**) Miyun Yuanwu's Dharma Lineage (4). (**e**) Miyun Yuanwu's Dharma Lineage (5). (**f**) Miyun Yuanwu's Dharma Lineage (6). (**g**) Miyun Yuanwu's Dharma Lineage (7).

The Feiyin Tongyong dharma lineage was also a thriving branch of the Miyun Yuanwu lineage with many disciples who were mainly active in Zhejiang but also in Fujian and Huguang. Feiyin Tongrong had a bhikkhuni disciple named Fajing Hao 法淨皓, and Tongrong's disciple Benchong Xingsheng 本充行盛 (d. 1671) had a bhikkuni disciple named Yizhen En 一真恩. Xuzhou Xingsheng 虛舟行省 (1600–1668) had a bhikkhuni disciple named Juewu 覺悟. Duguan Xingjing 獨冠行敬 (1613–1672) had a dharma heir named Shuijian Huihai 水鑑慧海, who had a bhikkhuni disciple named Shangji Ben 上機本. See Figure 5e.

Muyun Tongmen 牧雲通門 (1599–1671; from Jiaxing) had a bhikkhuni disciple named Yuanjian Xingxuan 圓鑑行玄 (1601–1673); her secular surname was Cao 曹, and she had published *yulu* 語錄 ('recorded sayings') and a *nianpu* 年譜 ('chronological biography'). Another of Muyun's disciples, who was named Zhongpu Xingru 中樸行如, had a bhikkhuni disciple named Foji Yuan 佛機元. See Figure 5f. Muyun's lineage developed in the Jiangsu–Zhejiang area.

Shanweng Daomin 山翁道忞 (1596–1674) had a bhikkhuni disciple named Jueke Xin 覺柯信, and Jueke Xin's dharma brother Lu'an Benyue 旅庵本月 (d. 1676) had a bhikkhuni disciple named Tianyi Ju 天一具. Yuan'an Benfeng 遠菴本僼 (1622–1682) had a disciple named Gulüfan 古律範, who had a bhikkhuni disciple named Ziyong Chengru 子雍成如 (b. 1648). See Figure 5g. Daomin's dharma lineage was most active propagating the dharma in the Jiangsu–Zhejiang area and also extended to the Huguang area.

Also, the dharma lineages of the Chan monastics of the Caodong denomination are as shown in Figure 6a. Compared with the development of the Linji denomination, the Caodong denomination was evidently not as big and powerful. Chan monastics of the Caodong denomination also had fewer works included in the Jiaxing Buddhist canon compared to the monastics of the Linji denomination. There are no records of bhikkhuni being included in Caodong dharma lineages. Even when consulting Caodong works, such as the *Wudeng xulüe* 五燈續略 (*An Abbreviated Continuation of the Five Genealogical Lamp Transmissions of Chan Buddhism*) by Yuanmen Zhu 遠門柱 (1602–1655), *Wudeng zuanxu* 五燈纘續 (*A Continuation of the Five Genealogical Lamp Transmissions*) by Hanyu Kuan 涵宇寬 (1596–1666), or even the 1672 *Zudeng datong* 祖燈大統 (*The Comprehensively Unified Lamps of the Progenitors*) by Jingfu 淨符, still no Chan bhikkhuni are to be found in the Caodong lineages of Chan. Despite this, the Caodong denomination has had an important influence on the development of Buddhism in the Ming–Qing period both in China and abroad (Chen 2020a, 2020b).

Again, in terms of the development of various branches of the Caodong denomination, Dajue Fangnian 大覺方念 (1552–1594) transmitted the dharma to Zhanran Yuancheng 湛然圓澄 (1561–1627) and Yuancheng transmitted the dharma to Sanyi Mingyu 三宜明盂 (1599–1655), Ruibai Mingxue 瑞白明雪 (1584–1641), Shiyu Mingfang 石雨明方 (1593–1648), Mailang Minghuai 麥浪明懷 (1586–1630), Ermi Mingfu 爾密明澓 (1590–1642), and others. Each of their disciples can be seen in Figure 6b. The extent of their dharma propagation was mostly in the Jiangsu–Zhejiang area, especially in the areas of Hangzhou and eastern Zhejiang. There are also traces of this lineage in Huguang, Jiangxi, and even Shanxi.

In addition, those who inherited the dharma lineage of Wuming Huijing 無明慧經 (1548–1618) include Shouchang Yuanmi 壽昌元謐 (1548–1618), Dayi Yuanlai 大艤元來 (1576–1630), Huitai Yuanjing 晦臺元鏡 (1577–1630), and Gushan Yuanxian 鼓山元賢 (1578–1657). Their respective inheritances can be seen in Figure 6c. This lineage propagated the dharma over a large area that included Jiangsu–Zhejiang, Jiangxi, Huguang, Fujian, and Guangdong. The members of this lineage were most numerous in the Guangdong–Fujian area.

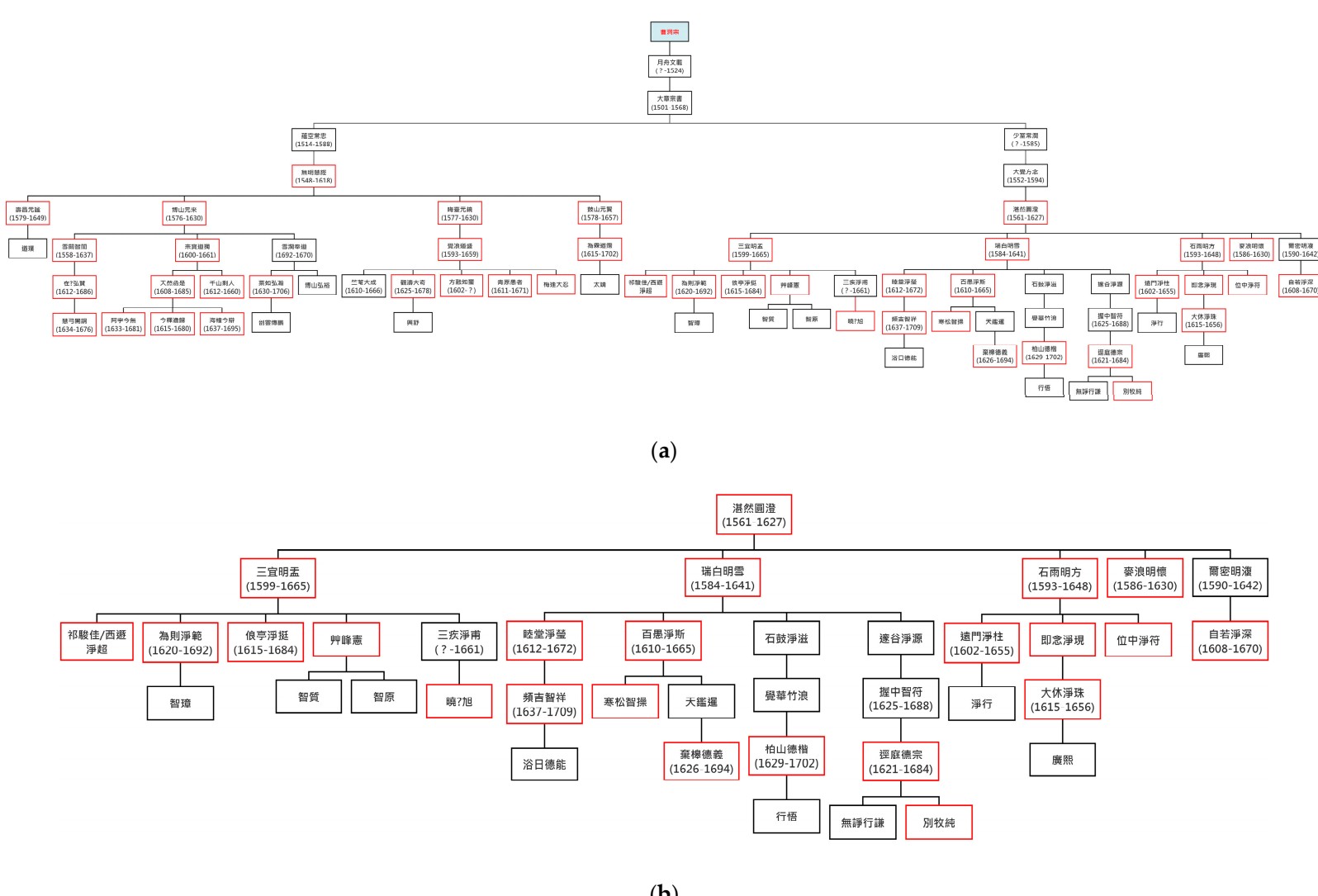

(**a**)

(**b**)

**Figure 6.** *Cont.*

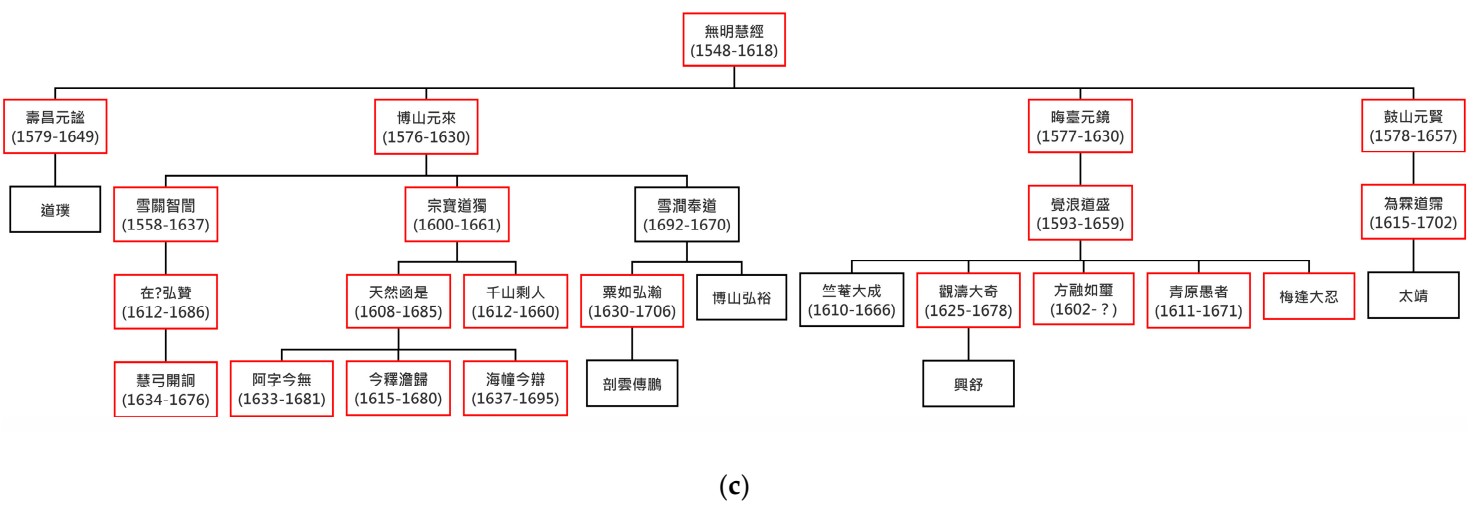

(**c**)

**Figure 6.** (**a**) The dharma lineages of the Caodong denomination in the late Ming and early Qing periods (1). (**b**) The dharma lineages of the Caodong denomination in the late Ming and early Qing periods (2). (**c**) The dharma lineages of the Caodong denomination in the late Ming and early Qing periods (3).

### 3. The Writings and Written Records of the Women Monastics of the Chan Dharma Lineage

The aforementioned is a summary of the dharma lineages of the Chan monastics whose works were included in the Jiaxing Buddhist canon. The greatest number of Chan bhikkhuni who were included in these dharma lineages, and whose works were included in the canon, were of the Linji denomination. Most of them belonged to the flourishing dharma lineage of Miyun Yuanwu. I have yet to find a single record of a Chan bhikkhuni being included in a Caodong dharma lineage, as shown in Figure 6a, and further efforts are needed. As seen in written records, there began to be more Chan bhikkhuni included in dharma lineage genealogies during the late Ming and early Qing periods, with some of them even having their recorded sayings included in the Jiaxing Buddhist canon.

Although men were still the primary rulers and leaders of society in the late Ming and early Qing, from the monastic lives, livelihoods, Buddhist practices, and dharma lineage inheritance of bhikkhuni of this period, together with what we can know about them from records of their families, the people of their localities, the cities they lived in, and the transitions of that time period, we can see that there was a gradual opening of society at this time. Huang Jingjia 黃敬家 says, 'Chan Buddhism held a relatively equal and open attitude towards women' (Huang 2004, p. 133). The recognition that Buddhist enlightenment is unrelated to gender provided bhikkhuni with more space for self-expression and activity. Although there was a great disparity between the proportions of men and women Chan monastics who could ascend the hall to give dharma talks or compile recorded sayings, the inclusion of the recorded sayings of women Chan monastics and their writings in the Jiaxing Buddhist canon can be considered a major breakthrough in Buddhism. This reflects the outstanding achievements both of these Chan bhikkhuni, as well as the support and assistance provided to these women by the open-minded men of their families, and of their local monastic and layperson communities. This support was a pivotal factor in allowing for the achievements of these women to be seen and passed down to posterity.

Based on this perspective, I hope in this article to further elaborate upon the instances of women Chan monastics being included in dharma lineages and having their works being published and passed down to posterity in the late Ming and early Qing, which have contributed to the construction of their personal status and their abilities to have a voice. This will allow for a more diverse, comprehensive, and accurate exploration of the factors involved.

In the late Ming and early Qing, the recorded sayings and works of women Chan monastics published in either the Jiaxing canon or the *Youxuzang* 又續藏 (*A New Supplement to the Buddhist Canon*) include the following:

1.  Jizong Xingche had her sayings compiled by Ruichu Chaoxing 瑞初超祥 in 1656 with the title *Jizong Che chanshi yulu* 季總徹禪師語錄 (*The Recorded Sayings of the Chan Master Jizong Che*).
2.  There are also recorded sayings of the bhikkhuni Zukui, preface dated 1670, entitled *Lingrui ni Zukui Fu chanshi Miaozhan lu* 靈瑞尼祖揆符禪師妙湛錄 (*The Miaozhan Cloister Records of the Chan Master Kuifu of Lingrui*), recorded by her student Zhao 焰 and others.
3.  Zukui also had a work together with Baochi Xuanzong, entitled *Songgu hexiang ji* 頌古合響集 (*The Concordant Sounds Collection of Verse Commentaries*),[3] also recorded by Zhao and others,
4.  The bhikkhuni Baochi Xuanzong also had her own published recorded sayings, preface dated 1673, entitled *Baochi Zong chanshi yulu* 寶持總禪師語錄 (*The Recorded Sayings of Chan Master Baochi Xuanzong*), recorded by her student Ming Ying 明英 and others.
5.  The bhikkhuni Chaochen 超琛 (Yikui 一揆; 1625–1679) had recorded sayings entitled *Cantong Yikui chanshi yulu* 參同一揆禪師語錄 (*The Recorded Sayings of Chan Master Cantong Yikui*), preface dated 1680, compiled by Puming 普明 and recorded by Mingjun 明俊.

6.    The bhikkhuni Diyuan Xinggang 衹園行剛 (1597–1654) also had her recorded sayings published under the title *Fushi Zhiyuan chanshi yulu* 伏獅衹園禪師語錄 (*The Recorded Sayings of Chan Master Zhiyuan of the Fushi Closter*), compiled by Shouyuan 授遠, Chaosu 超宿, and others.

7.    The bhikkhuni Yigong Chaoke 義公超珂 had her recorded sayings published in 1678 with the title *Fushi Yigong chanshi yulu* 伏獅義公禪師語錄 (*The Recorded Sayings of Chan Master Yigong of the Fushi Cloister*), recorded by Mingyuan 明元 and others.

According to my analysis, the reason why the works of these Chan bhikkhuni were included in the canon was, as mentioned before, closely related to their individual efforts and the advantageous conditions of their backgrounds, family connections, dharma connections, and regional connections. This also implies what Li Shi 李湜 pointed out in her *Ming-Qing guige huihua yanjiu* 明清閨閣繪畫研究 (*Research on Women Painters of the Ming-Qing Period*) on the importance of family connections as the key factor in whether a woman painter would be included in a history of painting or in biographies of painters. Li Shi writes:

> On whether a woman of the Ming–Qing period was fortunate enough to be included in a history of paintings, or in biographies of painters, the standard was 'not to value the inheritance of painting style, the appreciation and collection of their works, or their artistic achievements'; rather, it was their family background, such as whose woman, wife, or mother they were, and the circumstances of their romantic relationships with men that were regarded as important…The result of this was that some women who actually only barely proficient in painting, or who could not paint at all, were included in the records of such books only because of the particular circumstances of their family or of some romantic relationship…and the vast majority women painters who had no powerful man in behind them were rendered invisible without even the slightest mention (Li 2008, p. 10).

It is undeniable that the publication and inclusion of the aforementioned women Chan monastics' recorded sayings in the canon were not infrequently closely related to their family backgrounds. As shown in Figures 3 and 5a,c–g, the numbers of women Chan monastics included in the various dharma lineages of the Linji denomination varied. Generally speaking, they mainly belonged to two dharma lineages of Miyun Yuanwu 密雲圓悟 (1566–1642) and Yuanhu Miaoyong 鴛湖妙用 (1587–1642). In Miyun Yuanwu's dharma lineage, there were thirty-four women Chan monastics, with most being disciples of Biyuan Xinggang, who was herself a disciple of Shiche Tongcheng. Also, in the lineage of Jie'an Wujin 介菴悟進 (1612–1673), who was a disciple of Yuanhu Miaoyong, there were four Chan bhikkhuni. So, between these two lineages, there were a total of thirty-eight Chan bhikkhuni. As mentioned before, this is based on the analysis of the data I have collected so far and will be supplemented as I uncover additional relevant records.

Only seven of these thirty-eight women Chan monastics, or about eighteen percent of them, had their recorded sayings or works included in the Jiaxing canon. Nevertheless, this was a breakthrough in the development of Buddhism during the Ming–Qing period, and this highlights the openness of the privately printed Jiaxing canon. Even so, only a few of these bhikkhuni were able to be included in the canon, indicating that it was extremely rare for a bhikkhuni's works to be included. Some of these thirty-eight Chan bhikkhuni who did not have their recorded sayings included participated in the writing and compilation of the recorded sayings of other women Chan monastics whose works were included. Some of these bhikkhuni have published recorded sayings and other works that were not included in the canon, but these are quite few. Most of these women either had works that were not passed down or had no written works to be passed down.

The dharma lineages of these above-mentioned Chan bhikkhuni and the status of their writings are shown clearly in Table 2.

**Table 2.** Lineages of 38 female Chan masters and their writings.

| 1 | 2 | 3 | 4 | 5 | 6 |
|---|---|---|---|---|---|
| ◎Miyun Yuanwu 密雲圓悟 | ◎Shanweng Daomin 山翁道忞 | ◎Yuanan Benfeng 遠菴本懷 | Gulü Fan 古律範 | * (Nun) Ziyong Chengru 尼子雍成如 | Zuyuan 祖圓 |
| | | Lüan Benyue 旅庵本月 | (Nun) Tianyi Ju 尼天一具 | | |
| | | (Nun) Jueke Xin 尼覺柯信 | | | |
| | ◎Muyun Tongmen 牧雲通門 | Δ (Nun) Yuanjian Xingxuan 尼圓鑑行玄 | | | |
| | | Zhongpu Xingru 中樸行如 | (Nun) Foji Yuan 尼佛機元 | | |
| | ◎Feiyin Tongrong 費隱通容 | Duguan Xingjing 獨冠行敬 | ◎Shuijian Huihai 水鑑慧海 | (Nun) Shangji Ben 尼上機本 | |
| | | ◎Xuzhou Xingxing 虛舟行省 | (Nun) Juewu Zu 尼覺悟祖 | | |
| | | Benchong Xingsheng 本充行盛 | (Nun) Yizhen En 尼一真恩 | | |
| | | (Nun) Fajing Hao 尼法淨皓 | | | |
| | ◎Tongxuan Tongqi 通玄通奇 | Tianmu Chaozhi 天目超智 | (Nun) Zhaoqing Guang 尼照清光 | | |
| | | (Nun) Yinyue Xinglin 尼印月行霖 | | | |
| | | Fazhu Chang 法柱長 | Zizhi You 紫芝有 | (Nun) Tongli Jing 尼通禮敬 | |
| | ◎Fushi Tongxian 浮石通賢 | Fayin Xingcheng 法音行 | (Nun) Fumei Ji 尼桴梅濟 | | |
| | | (Nun) Gaoyuan Qing 尼高源清 | | | |
| | | Quanshi Wo 拳石沃 | (Nun) Huikong 尼慧空 | | |
| | ◎Hanyue Fazang 漢月法藏 | ◎Tuiweng Hongchu 退翁弘儲 | * (Nun) Lingrui Nizu 尼靈瑞尼祖 | | |
| | | | * (Nun) Baochi Xuanzong 尼寶持玄總 | | |
| | | | Boan Zhengzhi 檗庵正志 | (Nun) Daoyu 尼道遇 | |
| | | | (Nun) Renfeng Jiyin 尼仁風濟印 | | |
| | | Jude Hongli 具德弘禮 | Jubo Jiheng 巨渤濟恒 | (Nun) Lingxi Rong 尼靈璽融 | |
| | | | | (Nun) Huizhao Lian 尼慧照蓮 | |
| | | ◎Poushi Hongbi 剖石弘璧 | (Nun) Fayu Ying 尼法雨瀛 | | |
| | | Yimo Hongcheng 一默弘成 | Huotang Zhengyan 豁堂正喦 | (Nun) Xiang'an Hui 尼象菴慧 | |

**Table 2.** *Cont.*

| 1 | 2 | 3 | 4 | 5 | 6 |
|---|---|---|---|---|---|
| ◎Miyun Yuanwu 密雲圓悟 | ◎Shiqi Tongyun 石奇通雲 | Δ (Nun) Weiji Zhi 尼惟極致 | (Nun) Jingnuo Yue 尼靜諾越 | | |
| | Shiche Tongcheng 石車通乘 | * (Nun) Qiyuan Xinggang 尼祇園行剛 | ⊙(Nun) Yiran Chaosu 尼怡然超宿 | | |
| | | | ⊙(Nun) Puwen Shouyuan 尼普聞授遠 | | |
| | | | * (Nun) Yigong Chaoke 尼義公超珂 | | |
| | | | * (Nun) Yikui Chaochen 尼一揆超琛 | | |
| | | | Δ ⊙(Nun) Yiyin Chaojian 尼一音超見 | | |
| | | | (Nun) Yichuan Chaolang 尼義川超朗 | | |
| | | | (Nun) Guding Chaozhen 尼古鼎超振 | | |
| | ◎Longchi Tongwei 龍池通微 | * (Nun) Jizong Che 尼季總徹 | | | |
| ◎Yuanhu Miaoyong 鴛湖妙用 | ◎Jiean Wujin 介菴悟進 | ◎Shanduo Zhenzai 山鐸真在 | (Nun) Shengzhuo Di 尼聖拙地 | | |
| | | Suhong Zhenli 素弘真理 | Baoru Yu 寶如玉 | (Nun) Yungu Zong 尼蘊古宗 | |
| | | Lianhua Kedu 蓮花可度 | | | |
| | | (Nun) Mingxin Foyin 尼明心佛音 | | | |

Note: All female figures are marked by a backgrounded color of pink. ◎ Male Chan masters whose recorded sayings and other works were included in the *Jiaxing zang*. * Female Chan masters whose recorded sayings and other works were included in the *Jiaxing zang*. Δ Female Chan masters who were known to have recorded sayings or other works that were not included in the *Jiaxing zang*. ⊙ Nuns who assisted in recording or compiling the sayings of female Chan masters.

Table 2 shows the situation of the writings of thirty-eight women Chan monastics, with only a few having clear records. However, the lack of any recorded works being passed down does not indicate a lack of writing ability. But how to judge whether these Chan bhikkhuni were able to write? Although there is a lack of sufficient and direct written records, if we look at the backgrounds of these women and the bits and pieces after their ordinations, we might be able to get a general idea of the overall situation. Although not necessarily related to their writing abilities, it is possible to somewhat determine the contributing factors of whether a Chan bhikkhuni was included in dharma lineage genealogies by whether she had a family background that would have nurtured her literacy. We can consider this together with other factors, such as the social capital of her interpersonal relationships with other monastics and laypersons. These factors include her economic and cultural resources.

In my previous study, I concluded that according to the *Xu biqiuni zhuan*, there were sixty-two bhikkhuni with writing skills and works, such as poems, *gāthā*s, collected works, or recorded sayings, who were located in the southeastern seaport cities of China from the late Ming and early Qing periods before the Kangxi period. I added two more names I found in the women section (*guiyuan dian* 閨媛典) of the *Gujin tushu jicheng* 古今圖書集成 (*The Collection of Ancient and Modern Books and Illustrations*) for a total of sixty-four Chan bhikkhuni of this period who had writings that were passed down to posterity. Except

for a few who were born in the Fujian–Guangdong region, almost all, or about ninety percent, were born in the Jiangnan region. Their activities after becoming bhikkhuni were also mostly centered in Jiangnan seaport cities.[4] Benefiting from their convenient water transportation networks; flourishing humanistic scholarly atmospheres; thriving publishing networks that included the developed woodblock printing of bookstores as well as of families and private presses and a circulation of books from both within China and abroad; and prosperous commercial activities and maritime trade, seaport cities allowed for the Jiangnan region to have more favorable conditions in regard to there being many talented individuals and a great number of written works than compared to other regions. These pluralistically diverse and open seaport cities and their social resources indeed provided women with opportunities to show their achievements, be seen, and leave behind clear records of who they were and what they did.

Of the thirty-eight women Chan monastics mentioned before who were included in the dharma lineages of Chan, there is some overlap with the sixty-four bhikkhuni found between the *Xu biqiuni zhuan* and the women section of the *Gujin tushu jicheng*, but many of these sixty-four were not included in the dharma lineages of Chan. Clearly, it was not easy for women to be included in the Chan genealogies, to say nothing of the challenges of their works to be included in the Buddhist canon and disseminated. In order to further understand the possible circumstances whereby these thirty-eight women Chan monastics were included in Chan genealogies, I have organized some clues from their lives into Table 3.

**Table 3.** Lineage, background, and deeds of female Chan masters of late Ming and early Qing.

| Place of Birth/Abode | Buddhist Nun, Lineage Predecessor, and Successor | Family Background and Social Class (by Birth) | Writing Ability and Work | Provenance |
|---|---|---|---|---|
| Female Disciples in Shanweng Daomin's 山翁道忞 Lineage | | | | |
| 1. Hangzhou (native of Guandong 關東)/abbot of Biyun Yuan 碧雲院 Monastery in Hangzhou | Yongshou ni Ziyong 永壽尼子雍 (b. 1648), also known as Ziyong (ni) 子雍 (尼), Ziyong Ru 子雍如, Ziyong Chengru 子雍成如, Bixia Ziyong 碧霞子雍, Yongqing Ziyong 永慶子雍 Predecessor: Gulü 古律, 33rd patriarch of Linji Successors: Fozheng 佛證, Foliang 佛亮 | 荊門人氏, 祖籍關東, 流寓都門甚久. 父周諱志祥, 母牛氏. 開國之初, 父從駕屢著功勳, 不欲受官, 隱於耕讀, 樸素醇謹, 見善必為, 雅信釋氏. | *Ziyong Ru chanshi yulu* 子雍如禪師語錄 | *Ziyong Ru chanshi* yulu, *J.* no. B465, 39: 4.831b; *Wudeng quanshu* 107.664b; https://authority.dila.edu.tw/person/ (accessed on 16 June 2023; same below) |
| 2. Residing at Xiaoshan 蕭山 Nunnery in 吳興 | Tianyi Ju 天一具, Predecessor: Lüan Yue 旅庵月 (d. u) of Linji | Unknown | | *Xu biqiuni zhuan* 4.78; *Wudeng quanshu*, X 82: 539c; https://authority.dila.edu.tw/person/ |
| 3. Residing at Mingyin 明因 Nunnery in Zuili 檇李 | Jueke Xin 覺柯信, Predecessor: Muchen Min 木陳忞 (1596–1674) of Linji | Unknown | | *Xu biqiuni zhuan* 4.70; *Wudeng quanshu*, X 82: 380a; https://authority.dila.edu.tw/person/ |
| Female Disciples in Muyun Tongmen's 牧雲通門 (1599–1671) Lineage | | | | |
| 4. Changshu of Jiangsu/Zhizhi 直指 Nunnery in Yushan 虞山 | Yuanjian Xingxuan 圓鑑行玄 (1601–1673), also known as Yuanjian (Ni) 圓鑑(尼), Yuanjian Yuan 圓鑑元, Yuanjian Xuan 圓鑑玄 Predecessor: Muyun Tongmen 牧雲通門 of Linji | 常熟曹氏女. 幼出家, 謁報國受具, 嚴持律範. | *nianpu* 年譜, *yulu* 語錄 | *Xu biqiuni zhuan* 4.68; *Wudeng quanshu*, X 94.397b; https://authority.dila.edu.tw/person/ |

**Table 3.** *Cont.*

| Place of Birth/Abode | Buddhist Nun, Lineage Predecessor, and Successor | Family Background and Social Class (by Birth) | Writing Ability and Work | Provenance |
|---|---|---|---|---|
| 5. Dantu 丹徒/Secluding in Zehngju 正覺 Nunnery | Foji Yuan 佛機元 (d. u.) Predecessor: Helin Zhongpu Xingru 鶴林中樸行如 (d. u.) | Unknown | | *Xu biqiuni zhuan* 4.79; *Wudeng quanshu, X* 96.399c; https://author ity.dila.edu.tw/person/ |
| Female Disciples in Feiyin Tongrong's 費隱通容 Lineage | | | | |
| 6. Lushan 魯山/Yanshou 延壽 Nunnery in Jingzhou 荊州 | Shangji Ben 上機本 (d. u.) Predecessor: Shuijian Huihai 水鑑慧海 (1626–1687) of Linji | 魯山金氏女 | | *Xu biqiuni zhuan* 4.82; *Wudeng quanshu, X* 106; https://authority.dila.e du.tw/person/ |
| 7. Hangzhou/Qingliang 清涼 Nunnery | Juewu Zu 覺悟祖 (d. u.), also known as Miqi Chaochu 密啟超初 Predecessor: Xuzhou Xingxing 虛舟行省(1600–1668) of Linji | 杭之趙氏女 | | *Xu biqiuni zhuan* 4.76; *Wudeng quanshu, X* 926; https://authority.dila.e du.tw/person/ |
| 8. Shanyin 山陰/Taiping 太平 Nunnery in Wuhu 蕪湖 | Yizhen En 一真恩, also known as Taiping En太平恩 Predecessor: Benchong Xingsheng 本充行盛 (d. 1671) of Linji | 山陰王季重女也. | | *Xu biqiuni zhuan* 4.76; *Wudeng quanshu, X* 90. 650b; https://authority. dila.edu.tw/person/; https://digital.library. mcgill.ca/mingqing/ (明清婦女著作數據庫) (accessed on 16 June 2023; same below) |
| 9. Danghu 當湖/Sichan 思禪 Nunnery in Danghu | Fajing Hao 法淨皓 (d. u.), also known as Sichan Hao 思禪皓 Predecessor: Feiyin Tongrong 費隱通容 (1593–1661) of Linji | 當湖孫氏女, 父為水部臨鵡公, 母夢西域異僧投宿而生. 卯歲, 奇穎喜讀父書; 長則佐父出主蕪關稅政. 於歸未幾, 知世相匪堅, 乃請剃髮, 父許之, 遂投徑山費隱容披緇登具. | | *Xu biqiuni zhuan* 4.68; *Wudeng quanshu, X* 71. 353b; https://authority. dila.edu.tw/person/ |
| Female Disciples in Tongxuan Tongqi's 通玄通奇 Lineage | | | | |
| 10. Yongjia 永嘉/Baita 白塔 Nunnery in Yongjia | Qingzhao Guanghao 照清光皓 (d. u.) Predecessor: Tianmu Chaozhi 天目超智 (b. 1626) of Linji | 永嘉相國張文忠公孫女, 十三歲, 為父死難, 即持齋, 誓不出字. 二十四, 剃染于白塔, 參仙岩天目智和尚 (She could be the daughter of Zhang Xunye 張遜業.) | | *Xu biqiuni zhuan* 4.82; *Wudeng quanshu, X* 107.652a; https://author ity.dila.edu.tw/person/ |
| 11. Yaojiang 姚江/Fulong 伏龍 Nunnery | Yinyue Xinglin 印月行霖 (d. u.) Predecessor: Tiantong Tongqi 天童通奇(1596–1652) of Linji | 姚江黃太沖侄女. 自幼不染世緣, 生死心切, 常喜靜坐. 讀雲棲法匯, 宛如宿契, 遂動出塵之想. 十六, 適東山謝氏. 三載, 便改道妝, 進戒杭之理安. | *Fulong Yinyue chanshi yulu* 伏龍印月禪師語錄 | *Xu biqiuni zhuan* 5.85; *Wudeng quanshu, X* 79. 425a; https://authority.dila.e du.tw/person/CBDB (https://digital.library. mcgill.ca/mingqing/ (明清婦女著作數據庫) |

**Table 3.** *Cont.*

| Place of Birth/Abode | Buddhist Nun, Lineage Predecessor, and Successor | Family Background and Social Class (by Birth) | Writing Ability and Work | Provenance |
|---|---|---|---|---|
| 12. Ruian 瑞安/Shuangzhen 雙貞Nunnery in Ruian | Tongli Jing 通禮敬 (d. 1690) Predecessor: Zizhi You紫芝有 (d. u.) of Linji | 敏敬, 字通禮, 瑞安人, 俗姓陳, 乃林眉聲堂孀. | | *Xu biqiuni zhuan* 4.83; *Wudeng quanshu, X* 108.665a; https://authority.dila.edu.tw/person/ |
| Female Disciples in Fushi Tongxian's 浮石通賢 Lineage | | | | |
| 13. Zhenzhou 真州/Daci 大慈 Nunnery in Zhenzhou | Fuhai Ji 桴海濟 (d. u.) Predecessor: Fayin Xingcheng 法音行 (d. u.) of Linji Successors: Xinghua Foguo Guang 興化佛果廣, Foguo Hongchuan Guang 佛果弘傅廣, Daci Che 大慈徹 | Unknown | | *Xu biqiuni zhuan* 4.80; *Wudeng quanshu, X* 97.560b; https://authority.dila.edu.tw/person/ |
| 14. Shanyin of Shaoxing 紹興山陰/Puzhao 普照 Nunnery in Rugao 如皋 | Gaoyuan Qing 高源清 (d. u.) Predecessor: Fushi Tongxian 浮石通賢 (1593–1667) of Linji | 山陰金氏女. | | *Xu biqiuni zhuan* 4.71; *Wudeng quanshu, X* 77.408a; https://authority.dila.edu.tw/person/ |
| 15. Jin 鄞County/Xueluo 薛蘿 Nunnery in Jin County | Huikong 慧空 (d. u.) Predecessor: Quanshi Wo 拳石沃 (d. u.) of Linji | 鄞縣王氏女. | | *Xu biqiuni zhuan* 4.79; *Wudeng quanshu, X* 97.559b; https://authority.dila.edu.tw/person/ |
| Female Disciples in Hanyue Fazang's 漢月法藏 Lineage | | | | |
| 16. Huzhou of Zhejiang湖州/Lingrui 靈瑞 Nunnery in Huzhou | Zukui Xuanfu 祖揆玄符(d. u.), also known as Nun Zukui 祖揆, Jifu 濟符, Zukui Fu 祖揆符, Ni Zu Xuanfu 尼祖玄符, Ni Zukui Fu 尼祖揆符, Zufu 祖符, Lingrui Nizu 靈瑞尼祖 Predecessor: Tuiweng Chu 退翁儲禪師 (1605–1672, Lingyan 靈巖 of Suzhou) Successors: Shizhao 師炤, Yuelin 岳嶙, Zhenqing 振清, Zhenying 振渶, Zhencheng 振澂, Zhenhong 振鴻 | 濟苻, 字祖揆, 湖州李氏女. | *Zukui Fu chanshi yulu* 祖揆苻禪師語錄, *Lingrui Nizu Kuifu chanshi Miaozhan lu* 靈瑞尼祖揆符禪師妙湛錄 (5 *juan*), *Lingrui chanshi Yanhua ji* 靈瑞禪師喦華集 (5 *juan*), *Songgu hexiang ji* 頌古合響集 (1 *juan*, with co-author Baochi 寶持) | *Lingrui Nizu Kuifu chanshi Miaozhan lu, J* no. B338, vol. 35; *Lingrui chanshi Yanhua ji, J* no. B339, vol. 35; *Xu biqiuni zhuan* 4.74; *Wudeng quanshu, X* 87.479a; Su, 'Nüchan hexiang'; https://authority.dila.edu.tw/person/ |
| 17. Jiaxing 嘉興/Miaozhan 妙湛Nunnery in Jiaxing | Baochi Xuanzong 寶持玄總 (d. u.) also known as Baochi (ni) 寶持(尼), Baochi Zong 寶持總, Baochi Xuanzong 寶持玄總, Jin Shuxiu 金淑脩 Predecessor: Tuiweng Chu 退翁儲 (1605–1672) of Linji | 嘉興金氏女, 隨州太僕之家媳. 夫亡, 脫俗於妙湛, 力參有省. | *Baochi Zong chanshi yulu (ni)* 寶持總禪師語錄 (尼), *Songgu hexiang ji* 頌古合響集 (1 *juan*, with co-author Xuanfu 玄符) | 'Guixiu Jin Shuxiu' 閨秀金淑脩 ('Gentlewoman Jin Shuxiu'), in *Guochao huazheng xulu, juan* 2; *Guochao guixiu zhengshi xuji, juan* 2; *Xu biqiuni zhuan* 4.75; *Wudeng quanshu, X* 87.479a; *Runxiu Jinshu xiu*; https://authority.dila.edu.tw/person/ |

**Table 3.** *Cont.*

| Place of Birth/Abode | Buddhist Nun, Lineage Predecessor, and Successor | Family Background and Social Class (by Birth) | Writing Ability and Work | Provenance |
|---|---|---|---|---|
| 18. Residing at Longhu 龍護 Nunnery in Wuxi 無錫 | Daoyu 道遇 (d. u.) Predecessor: Boan Zhengzhi 檗庵正志 (1600–1676) of Linji | Unknown | | *Xu biqiuni zhuan* 4.82; *Wudeng quanshu* 105.631b; https://authority.dila.edu.tw/person/ |
| 19. Kunshan 崑山/Lingzhi 靈峙 Nunnery in Yufeng 玉峰 | Renfeng Jiyin 仁風濟印 (d. u.) Predecessor: Tuiweng Honghchu 退翁弘儲 (1605–1672) of Linji | 崑山顧文康公 (顧鼎臣; 1473–1540) 從孫女. | *Jiyin Renfeng yulu* 濟印仁風語錄, (1 *juan*) | *Xu biqiuni zhuan* 4.74–75; *Wudeng quanshu* 87.479b; https://authority.dila.edu.tw/person/ |
| 20. Residing at Nianhua 拈花 Nunnery in Jiangdu 江都 | Lingxi Rong 靈璽融 (d. u.), also known as Nianhua Lingxi 拈花靈璽, Huarong 花融 Predecessor: Jubo Jiheng 巨渤濟恆 (1605–1666) of Linji | 胎素天秉, 剃染於拈花庵. | | *Xu biqiuni zhuan* 4.81; *Wudeng quanshu* 103.621; *Yushan heshnag yulu*, J 40:19.608; https://authority.dila.edu.tw/person/ |
| 21. Jinsha 金沙/Gudi 古滌 Nunnery in Suzhou | Huizhao Lian 慧照蓮 (d. u.), also known as Gudi Lian 古滌蓮 Predecessor: Jubo Jiheng of Linji | 金沙孫氏女, 年十九, 白父出家於蘇州天池 古滌庵剃染, 謁揚州天寧巨渤恒公, 言下知歸, 得蒙印可. | | *Xu biqiuni zhuan* 4.81; *Wudeng quanshu* 103.621c; https://authority.dila.edu.tw/person/ |
| 22. Chongming 崇明/Zideng 自登 Nunnery in Chongming | Fayu Ying 法雨瀛 (d. u.), also known as Zideng Ying 自登瀛 Predecessor: Poushi Bi 剖石璧 (1599–1670) of Linji | 崇明管氏女. 年二十八, 出家自登. | | *Xu biqiuni zhuan* 4.73; *Wudeng quanshu* 83.455c; https://authority.dila.edu.tw/person/ |
| 23. Hangzhou/Yun'an 筠菴 Nunnery in Wuling 武陵 | Xiang'an Hui 象菴慧 (d. u.) Predecessor: Huotang Zhengyan 豁堂正喦 (1597–1670) | 武林莊氏女. | | *Xu biqiuni zhuan* 4.80; *Wudeng quanshu* 103.619a; https://authority.dila.edu.tw/person/ |
| Female Disciples in Shiqi Tongyun's 石奇通雲 Lineage | | | | |
| 24. Yao-jiang 姚江/Xiongsheng 雄聖 Nunnery in Hangzhou | Weiji Zhi 惟極致 (d. 1672), also known as Weiji (ni) 惟極(尼), Xiongsheng Zhi 雄聖致, Xiongsheng Xingzhi 雄聖行致, Weiji Xingzhi 惟極行致 Predecessor: Shiqi Tongyun 石奇通雲 (1594–1663) of Linji Successor: Jingnuo Yue 靜諾越 | 姚江名家女. 童真入道, 常隨父參密雲悟於天童, 復參石奇雲於雪竇. | *Xiongsheng Weiji chanshi yulu* 雄聖惟極禪師語錄 (3 *juan*) | *Xu biqiuni zhuan* 4.90; *Wudeng quanshu* 95.389b; https://authority.dila.edu.tw/person/ |
| 25. Hangzhou/Xiongsheng Nunnery in Hangzhou | Jingnuo Yue 靜諾越 (d. u.) Predecessor: Weiji Zhi of Linji | 武林林氏女. 幼出家於雄聖庵, 依惟極致為師, 往來雪竇之門. | | *Xu biqiuni zhuan* 4.78; *Wudeng quanshu* 95; https://authority.dila.edu.tw/person/ |

**Table 3.** *Cont.*

| Place of Birth/Abode | Buddhist Nun, Lineage Predecessor, and Successor | Family Background and Social Class (by Birth) | Writing Ability and Work | Provenance |
|---|---|---|---|---|
| Female Disciples in Shiche Tongcheng' s 石車通乘 Lineage | | | | |
| 26. Jiaxing/Hu'an 胡庵 Nunnery, Fushi Monastery 伏獅 in Meixi 梅溪 | Qiyuan Xinggang 祇園行剛 (1597–1654), also known as Qiyuan (ni) 祇園(尼), Qiyuan Gang 祇園剛, Fushi Qiyuan 伏獅祇園, Fushi Qiyuan Gang伏獅祇園剛, Fushi Gang 伏獅剛 Predecessor: Jinsu Shiche Tongcheng 金粟石車通乘 (1598–1638) of Jiaxing Successors: Chaojian 超見, Chaochen 超琛, Chaoke 超珂, Puwen Shouyuan 普聞授遠, Yiran Chaosu 怡然超宿, Yichuan Chaolang 義川超朗, Guding Chaozhen 古鼎超振, Baochi Chaozhan 寶持超湛, Langyue Mingnei 朗月明內, Yuanyan Tongmeng 雲巖通猛, Yingjue Chaogui 穎覺超珪, Chaoyi 超義, Chaoyin 超蔭, Chaohui 超慧 | 處士胡日華女. 幼有至性, 好禪靜. 父母不聽所願, 歸於諸生常公振. 未幾而寡, 茹素奉佛, 誓欲了生死. | *Qiyuan Gang chanshi yulu* 祇園剛禪師語錄 | *Jiaxing fuzhi*, juan 62; *Wanqing yi shihui, juan* 199; *Fushi Yigong chanshi yulu, J* no. B435, 39: 2.437, 439; *Xu biqiuni zhuan* 4.60; *Wudeng quanshu* 72.354a; https://authority.dila.edu.tw/person/ |
| 27. Unknown | Yiran Chaosu 怡然超宿(d. u.) Predecessor: Qiyuan Xinggang of Linji | Unknown | *Fushi Qiyuan chanshi yulu* 伏獅祇園禪師語錄 (comp. Shouyuan 授遠, Chaolang 超朗, Chaoke 超珂, Chaojian 超見, Chaozhen 超振, Chaochen 超琛) | *Fushi Yigong chanshi yulu, J* no. B435, 39: 2.439; https://authority.dila.edu.w/person/ |
| 28. Unknown | Puwen Shouyuan 普聞授遠 (d. u.) Predecessor: Qiyuan Xinggang of Linji | Unknown | Ibid. | *Fushi Yigong chanshi yulu, J* no. B435, 39: 2.439; https://authority.dila.edu.w/person/ |
| 29. Jiaxing/Fushi Monastery in Meixi 梅溪, Bore 般若 Nunnery in Nanxun 南潯 | Yigong Chaoke 義公超珂 (1614–1661), also known as Yigong (ni) 義公(尼), Kuiying 揆英, Yigong Ke 義公珂, Fushi Yigong 伏獅義公 Predecessor: Qiyuan Xinggang of Linji | 出身名閨巨族. | *Fushi Yigong chanshi yulu* 伏獅義公禪師語錄 | *Fushi Yigong chanshi yulu, J* no. B435, 39: 1, 5; *Fushi Zhiyuan chanshi yulu, J* no. B210, 28: 2.439; https://authority.dila.edu.w/person/ |

**Table 3.** *Cont.*

| Place of Birth/Abode | Buddhist Nun, Lineage Predecessor, and Successor | Family Background and Social Class (by Birth) | Writing Ability and Work | Provenance |
|---|---|---|---|---|
| 30. Jiaxing/Cantong 參同 Nunnery in Jiaxing | Yikui Chaochen 一揆超琛 (1625–1679), also known as Yikui (ni) 一揆(尼), Yikui Chen 一揆琛, Cantong Yikui 參同一揆 Predecessor: Qiyuan Xinggang of Linji Successors: Puming 普明, Mingjun 明俊 | 嘉興大司寇孫簡蕭公之曾孫女. 父孫茂時, 母高氏超臻, 長兄孫子彰、仲兄子麟(鍾瑞), 少年得道, 登三教壇. 琛幼聰敏, 不由師傅, 而通書義, 兼善繪墨. | *Cantong Yigui chanshi yulu* 參同一揆禪師語錄 | *Xu biqiuni zhuan* 4.77; *Cantong Yigui chanshi xingshi*, in *Cantong Yikui chanshi yulu*, J no. B436, 39: 16a; *Fushi Zhiyuan chanshi yulu*, *juan* 2; *Wudeng quanshu* 93.552c; https://authority.dila.edu.tw/person/ |
| 31. Residing in Hu'an Nunnery and Fushi Monastery in Meixi | Yiyin Chaojian 一音超見 (d. u.) Predecessor: Qiyuan Xinggang of Linji | 姓戈氏. | *Taishan pozi song* 台山婆子頌 | *Xu biqiuni zhuan* 4.76; *Wudeng quanshu* 93.517c; https://authority.dila.edu.tw/person/ |
| 32. Unknown | Yichuan Chaolang 義川超朗 (d. u.) Predecessor: Qiyuan Xinggang of Linji | Unknown | *Fushi Qiyuan chanshi yulu* | *Fushi Zhiyuan chanshi yulu*, J no. B210, 28: 2.439; https://authority.dila.edu.w/person/ |
| 33. Unknown | Guding Chaozhen 古鼎超振 (d. u.) Predecessor: Qiyuan Xinggang of Linji | Unknown | Ibid. | *Fushi Zhiyuan chanshi yulu*, J no. B210, 28: 2.439; https://authority.dila.edu.w/person/ |
| Female Disciples in Longchi Tongwei's 龍池通微 Lineage | | | | |
| 34. Hengzhou 衡州/Huideng 慧燈 Monastery in Suzhou, Pudu 普度 Nunnery in Xinghua 興化, Shanhu 善護 Nunnery in Danghu 當湖 | Xingche 醒徹, (stylename: Jizong 季總; b. 1606), also known as Xingche 行徹, Jizong (ni) 季總(尼), Jizong Che 季總徹, Mingche 明徹, Jizong Che 繼總徹 Predecessor: Longchi Wei 龍池微 (1594–1657) | 劉氏女, 父劉善長. 好閱儒書、佛經, 痛念生死, 厭處塵凡. | *Jizong Che chanshi yulu* 季總徹禪師語錄 (2 *juan*) | *Jizong Che chanshi yulu*, J no. B211, 28: 1.442; 2.453; *Xu biqiuni zhuan* 4.66; *Wudeng quanshu*, *juan* 72; https://authority.dila.edu.tw/person/ |
| Female Disciples in Jiean Wujin's 介菴悟進 Lineage (1612–1673) (Heir to Yuanhu Miaoyong 鴛湖妙用 (1587–1642)) | | | | |
| 35. Residing at Zengfu 增福 Nunnery in Qishui 蘄水 | Shengzhuo Di 聖拙地 (d. u.) Predecessor: Shanduo Zhenzai 山鐸真在 (1621–1672) | Unknown | | *Xu biqiuni zhuan* 4.80; *Wudeng quanshu* 102.609a; https://authority.dila.edu.tw/person/ |
| 36. Unknown | Yungu Zong 蘊古宗 (d. u.) Predecessor: Baoru Yu 寶如玉 (d. u.) | Unknown | | *Wudeng quanshu* 108.670c; https://authority.dila.edu.tw/person/ |
| 37. Huaian 淮安/Lianhua 蓮花 Nunnery in Xiuzhou 秀州 | Lianhua Kedu 蓮花可度 (d. u.) Predecessor: Jiean Wujin 介庵悟進 (1612–1673) of Linji | 淮安田氏季子. 父官, 以指揮罪而歿. 度方七歲, 見父屍感歎, 便有出塵志. | | *Xu biqiuni zhuan* 4.72; *Wudeng quanshu* 81.447b; https://authority.dila.edu.tw/person/ |

**Table 3.** *Cont.*

| Place of Birth/Abode | Buddhist Nun, Lineage Predecessor, and Successor | Family Background and Social Class (by Birth) | Writing Ability and Work | Provenance |
|---|---|---|---|---|
| 38. Puzhen 濮鎮 of Zuili 檇李/Mingyin 明因 Nunnery in Zuili | Mingxin Foyin 明心佛音 (d. 1674) Predecessor: Jiean Wujin of Linji | 檇李濮鎮葉氏女. 幼喪母, 延僧誦經, 見地獄畫相, 即心動不茹葷, 矢志出家. 父為締姻, 死誓不從, 遂投明因庵依師落髮, 苦行數載. | | *Xu biqiuni zhuan* 4.73; *Wudeng quanshu* 81.447b; https://author ity.dila.edu.tw/person/ |

Table 3 shows that for many of these women, there is still a lack of clarity regarding the family backgrounds and the circumstances of their written works, while we have comparatively detailed information about the lives of seven of them—here numbered 1, 16, 17, 26, 29, 30, and 34; these are the same women who had their recorded sayings included in the Jiaxing canon. Details regarding the family backgrounds, reasons for monastic ordination, and the support from monks and laypersons as well as other social support that facilitated the publication and inclusion of their recorded sayings in the canon for these seven are analyzed more specially in one of my previous studies:

> Looking closely at these seven women Chan monastics whose recorded sayings were included in the canon, we find that, except for the bhikkhuni Jizong Xingche and Ziyong Chengru, most of all these women were born in seaport cities of the Jiangnan region, especially Jiaxing, and all of them were from eminent literati families or the families of officials. It was not just these women Chan monastics themselves; the famous officials and gentry who supported them and those who wrote prefaces to their recorded sayings or encouraged their publication were also mostly from Jiaxing or other cities around the Jiangnan region. It can be seen that a combination of geographical location, family connections, and dharma connections was an essential prerequisite for these few women Chan monastics to have their recorded sayings published and included in the Buddhist canon to be disseminated (Chen 2022, particularly Part 6).

Number 24, Weiji Xingzhi 惟極行致 (d. 1672), was a dharma heir of Shiqi Tongyun 石奇通雲 (1594–1663). She authored her *Xiongsheng Weizhi Chanshi yulu* 雄聖惟極禪師語錄 (*The Recorded Sayings of Chan Master Xiongsheng Weiji*) in three fascicles, but her text was not included in the Jiaxing canon. And although numbers 27, 28, 29, 32, and 33 were not included in the transmission genealogy of the *Wudeng quanshu*, number 29, Yigong Chaoke 義公超珂 (1614–1661), had her recorded sayings *Fushi Yigong Chanshi yulu* 伏獅義公禪師語錄 (*The Recorded Sayings of Chan Master Yigong of Fushi Cloister*) included in the Jiaxing canon. At the same time, it is clearly stated in the *Fushi Qiyuan chanshi yulu* 伏獅祇園禪師語錄 (*The Recorded Sayings of Chan Master Qiyuan of the Fushi Cloister*) that Yigong Chaoke was a dharma heir disciple of Qiyuan Xinggang (1597–1654) of the thirty-seventh generation of the Linji dharma-lamp transmission; Yigong Chaoke received recognition of dharma transmission within a community of women Chan monastics, and so, this study clearly indicates the lineage of her dharma transmission inheritance (Su 2003, p. 24; 2008; Guo 2010). There are five women Chan monastics mentioned in the recorded sayings of Qiyuan Xinggan, all of whom were Qiyuan Xinggan's disciples who were also writers; most of them were engaged in recording and compiling Qiyuan Xinggan's recorded sayings, as shown in Table 3.

The family backgrounds are unclear for the women I have numbered 2, 3, 5, 13, 18, 35, and 36. For the others, there are mostly only the following few details about them. Number 4, Xingxuan Yuanjian 行玄圓鑑, was a woman of the Cao 曹 family of Changshu 常熟 (Jiangsu). Number 6, Shangji Ben 上機本, was a woman of the Jin 金 family from Lushan

魯山 (Henan). Number 7, Juewu 覺悟, was a woman of the Zhao 趙 family of Hangzhou
杭州 (Zhejiang). Number 12, Tongli Jing 通禮敬 (d. 1690), was originally surnamed Chen
陳 and was aunt-in-law of Lin Meisheng 林眉聲 (i.e., Li Qihong 林齊鋐 from Rui'an 瑞安,
Zhejiang). Number 14, Gaoyuan Qing 高源清, was a woman of the Jin 金 family of Shanyin
山陰 (an ancient name for Shaoxing, Zhejiang). Number 15, Huikong 慧空, was a woman
of the Wang 王 family of the Yin 鄞 prefect (Zhejiang). Number 21, Huizhao Lian 慧照蓮,
was a woman of the Sun 孫 family of Jinsha 金沙 (Jiangsu). Number 22, Fayu Ying, 法雨瀛,
was a woman of the Guan 管 family of Chongming 崇明 (Zhejiang). Number 23, Xiang'an
Hui 象菴慧, was a woman of the Zhuang 莊 family of Wulin 武林. Most descriptions only
go so far. With such limited information, it is necessary to search for possible indirect clues
by inference to supplement what we can know about the identity and doings of some of
these Chan bhikkhuni.

**4. From Daughters of Eminent Families to Bhikkhuni of the Chan Lineage**

As mentioned earlier, the identity and life events of quite a few of these women Chan
monastics are unclear, but by exploring possible clues, we might be able to gain a little
more clarity. Take number 4, Yuanjian Xingxuan, for example. There is an entry for her in
the sixth fascicle of the *Zhengyuan lüeji* 正源略集 (*Collected Outline of the Orthodox Source*'),
entitled 'Yushan zhizhi ni Yuanjian Xuan Chanshi' 虞山直指尼圓鑑玄禪師 ('The Directly-
pointing (to enlightenment) Chan Master bhikkhuni Yuanjian Xuan of Mount Yu'):

> A woman of the Cao family of the city [of Changshu 常熟], Yuanjian renounced
> family life when she was a child, and she went to Baoguo to receive full monastic
> precepts. She strictly observed the precepts. At first, Yuanjian visited Miyuan
> Yuanwu, and after paying her respects to him on several occasions, she raised
> a fist. Miyuan then hit her with his stick and said, 'What's this?' Yuanjian
> said, 'A thousand sages wouldn't understand'. Miyuan said, 'Let it go'. She
> said, 'Let go of what?' Miyuan then left. Later, Yuanjian went to Muyun Tong-
> men to receive the dharma, and she went into seclusion on Mount Yu. One
> day, Huang Chunyao, presented literatus [*jinshi*], came to her gate and inquired
> of her, saying, 'I have long admired this gatekeeper'. Yuanjian said, 'I who
> am lacking in the Way will never understand'. The literatus said, 'As soon as
> you open your mouth, there's a tangled mess of the vines of language; I want
> you to show it all to me'. Yuanjian said, 'Let it go'. The literatus was over-
> joyed. 邑之曹氏女. 幼出家, 詣報國受具, 嚴持律範. 初參金粟悟. 禮拜次,即竪拳.
> 悟便棒曰, '者箇是甚麼'? 師曰, '千聖不識'. 悟曰, '放下著'. 師曰, '放下箇什麼'?
> 悟乃休. 後謁古南得法, 掩關虞山. 黃淳耀進士一日到關, 問曰, '久慕關主'. 師曰,
> '貧道總不識'. 士曰, '啟口即是葛藤, 要師全提'. 師曰, '放下著'. 士大悅 (Jiyuan,
> Liaozhen, and Dazhen 1975–1989, 6:40a23–b4).

According to the *Mingshi* 明史 (*History of the Ming*), the literatus of this story, Huang
Chunyao 黃淳耀 (1605–1645), was from Jiading 嘉定. His courtesy name (*zi* 字) was Yun-
sheng 蘊生. He was awarded the *jinshi* degree in the sixteenth year of the Chongzhen 崇禎
reign (1643). When the capital of the Ming fell to the Manchus and the Prince of Fu 福王
established a southern capital, all the *jinshi* went to be appointed as officials except for
Chunyao, who did not go to be considered for a position. When the southern capital of
the Ming also fell to the Manchus, as well as Jiading, Chunyao ended his own life together
with his younger brother Yuanyao 淵耀. This was on the twenty-fourth day of the seventh
month of the Hongguang reign (1645). Chunyao's writings include the fifteen fascicles of
the *Tao'an ji* 陶菴集 (*Tao'an Collection*). His followers gave him the posthumous title Zhen-
wen 貞文.[5]

When Jiading was massacred, it was recorded:

> In this battle, more than twenty thousand people died inside and outside the
> city. Among the literati were Hou Dongchang, Huang Chunyao, [and] Gong
> Yongyuan. Among those who had been successful candidates in the provincial

civil service examination (*xiaolian* 孝廉) was Zhang Ximei. Among those who had been successful candidates in the national civil service examination (*gong-shi* 貢士) was Wang Yuncheng. Among those who had been successful candidates in the county civil service examination (*qingyi* 青衿) were Huang Yuanyao, Hou Yuanyan, Hou Yuanjie, and seventy-eight others. At that time, a countless number of filial sons and grandsons, chaste husbands and loyal wives, talented men, and beautiful women were killed by arrow or blade. This unforeseen calamity was unprecedented in the entire history of the district. 是役也, 城內外死者二萬餘人. 搢紳則侯峒曾、黃淳耀、龔用圓,孝廉則張錫眉, 貢士則王雲程, 青衿則黃淵耀、侯元演、侯元潔等七十八人. 其時孝子慈孫、貞夫烈婦、才子佳人, 橫罹鋒鏑者不計其數. 謂非設縣以來絕無僅有之異變也哉.[6]

Huang Chunyao and many other scholars died during the massacre. According to the postscript by Bao Tizhai 鮑體齋 in the *Tao'an shiji* 陶庵詩集 (*Tao'an Poetry Collection*):

> The foundations of Huang Chunyao's learning were Chan Buddhism, and from the beginning, this is never something he denied. His brilliant discussions all got to the meaning of Chan, and it is not at all that he was using Chan to clarify Confucianism. […] Of all of the records of my teacher that I have compiled, the majority are the sharp hits and shouts of the Chan school. 陶庵先生之學, 本於禪宗, 亦初不自諱. 其鑿鑿諸議論者, 皆禪門宗旨, 併非藉禪以明儒. ……所輯吾師錄, 亦大半落禪家棒喝機鋒.

We can see that Huang Chunyao liked to inquire into Chan in his daily life. In his poetry collection, there are poems about Mount Yu, such as 'Yushan chi hurou zuo' 虞山喫虎肉作 ('Eating Tiger Meat on Mount Yu') (Huang 1676, 4:11–12) and 'Jiuri deng Yushan yuyu su Xingfu Chanyuan ershou' 九日登虞山遇雨宿興福禪院二首 ('Two Poems on Getting Stuck in the Rain and Staying at the Xingfu Chan Monastery while Climbing Mount Yu for the Double Ninth'). Although it is unclear which year Huang Chunyao climbed Mount Yu, in the first of these two poems is written:

> The tranquil hue of the abyss; this emptiness is always present day or night.
>
> 潭影悠悠日夜空
>
> A night of thunder and rain; the haze of primordial chaos.
>
> 一宵雷雨氣鴻濛
>
> Sipping tea and talking together about how nothing is born or dies.
>
> 啜茶共說無生話
>
> Paying full respects and still looking up to the preeminent hero. (The monks took out an image of Zibo Zhenke 僧出紫柏老人像). (Huang 1676, 6:11–12)
>
> 禮足還瞻蓋世雄

Zibo Zhenke 紫柏真可 (1543–1603), who initiated the printing of the Jiaxing Buddhist canon, was clearly venerated at the Xingfu Chan Monastery on Mount Yu and by the monks who lived here. Certainly, Huang Chunyao must have also known about Zibo Zhenke and his work to publish a Buddhist canon. The publication of this Buddhist canon was major news around China and especially for the literati of the Jiangnan region who contributed a considerable amount of labor and material resources. Although seemingly unrelated, perhaps this occasion when Huang Chunyao's climbed Mount Yu for the Double Ninth Festival of an unspecified year was the same time he knocked at the gate of Yuanjian Xingxuan during her period of secluded practice on that mountain.

The aforementioned entry of Yuanjian Xingxuan continues:

Qian Qianyi's 錢宗伯 (i.e., Qian Qianyi 錢謙益 [1582–1664]) wife asked, 'What is the return of all phenomena to oneness, and where does oneness return?' Yuanjian said, 'The beautiful waters are beautiful year after year, and the verdant hills are verdant year after year'. Yuanjian was not ill on the eleventh day of the second

month of the *guichou* year of the Kangxi reign (1673), but she suddenly ordered her head to be shaved. She took a bath and changed her robes. Then she sat in the lotus position and uttered the following *gāthā*: 錢宗伯夫人問, '如何是萬法歸一, 一歸何處'? 師曰, '秀水年年秀, 青山歲歲青'. 康熙癸丑二月十一日, 無疾, 忽命剃頭, 沐浴更衣, 趺坐說偈曰:

For seventy-three years I have borrowed false names,

七十三年假借名

and not a single thing could be agreeable to me.

了無一法可當情

Now the four elements [of my body] are dispersing in accordance with their functioning,

而今四大隨機散

I have already said goodbye to the donors.

曾向檀那致別聲

Then she looked happy as she died. She has recorded sayings and a chronological biography that are circulating in the world. 遂怡然而化. 《語錄》、《年譜》行世 (Jiyuan, Liaozhen, and Dazhen 1975–1989, 6:40b5–9).

Qian Qianyi was a famous scholar of the late Ming and early Qing periods; his wife here was probably Liu Rushi 柳如是 (1618–1664). Qian Qianyi had close contact with Buddhist monastics and laypersons in various places throughout the Jiangnan region in the late Ming and early Qing periods. He also claimed to be a lay disciple of Zibo Zhenke, and he enthusiastically assisted in the later stages of the publication of the Jiaxing Buddhist canon with great effort.[7]

At this point, although the family background of Yuanjian Xingxuan is still unclear, from her interactions with a famous literati in the Jiangnan region, her birth in Changshu, her activities on Mount Yu of Changshu, that her teacher Muyun Tongmen 牧雲通門 (1599–1677) was also from Changshu, and that she transmitted the dharma in seaport areas such as Jiaxing, we can see that she was deeply supported in her whole life of Buddhist practice and by her regional connections. Of course, Yuanjian Xingxuan's inclusion in the dharma lineage of Chan was due in no small part to her teacher's recognition of her outstanding achievements in personal cultivation. She also had recorded sayings and a chronological biography that she left to the world, but unfortunately, these publications are no longer extant. We can see that Yuanjian Xingxuan was an abundantly talented and perceptive woman who stood out for her refinement.

Number 7 is Juewu 覺悟, also known as Miqi Chaochu 密啟超初. Juewu was in the dharma lineage of Xuzhou Xingsheng 虛舟行省 (1600–1668). The *Wudeng quanshu* writes that she was 'a woman of the Zhao family of Hangzhou' 杭之趙氏女. I was unable to find the exact identity of Juewu's secular family, but in the recorded sayings of Xuzhou Xingsheng, there is a verse, 'Yu Juewu shangzuo (gengzi Miqi)' 與覺悟上座 (更字密啟) ('Presenting to Elder Juewu (changing her courtesy name [*zi* 字] to Miqi 密啟')), in which is written:

You came to Nanping

君到南屏來

Like a wonderful friend who saves me even before I ask for help.

儼然不請友

I was ashamed I had no pure offerings,

予愧無清供

I delighted only to silently observe.

所喜惟澹守



You ran around to Chan temples here and there looking for a popular place,

趨闤走門庭

You mixed up the yellow bell for the earthenware-pot drum.

黃鐘雜瓦缶

Since you haven't yet recognized the tiger of the Buddha,

未識大雄虎

How will you avoid Zhaozhou's dog?

寧避趙州狗

Understanding the dharma is knowing cautiousness,

識法知慎重

Never act carelessly.

操履端不苟

You can understand my idea,

君能會予意

A relationship can only last if it's extremely intimate. (Xuzhou 1987, 2:375c)

莫逆方堪久

From this, we can know that Juewu (Miqi Chaochu) based her Buddhist practice somewhere in the mountains to the south of West Lake in Hangzhou. We can also see that she had a close relationship with Xuzhou Xingsheng, who highly valued and approved of her, and that she was no ordinary person.

Number 8, Yizhen En 一真恩 (also known as En of Taiping Convent 太平恩), was a daughter of Wang Jizhong 王季重 (1574–1646) of Shaoxing. Wang Jizhong's given name was Siren 思任, Jizhong 季重 was his courtesy name, and his alternative name was Suidong 遂東. Wang Jizhong was an eminent literatus of Shaoxin, who was in frequent contact with other famous persons and literati of the Jiangnan region; he was good at poetry, calligraphy, and painting. He had eight sons, and three daughters named Wang Jingshu 王靜淑, Wang Duanshu 王端淑, and Wang Zhenshu 王貞淑. Wang held the highest regard for his second daughter, Wang Yuying 王玉瑛, whose given name was Duanshu 端淑. Deng Hanyi 鄧漢儀 (1617–1689) described Wang Duanshu as follows:

> Wang Jizhong of Shanyin had eight sons, but only his daughter Yuying followed in her father's footsteps by using her talents to be skilled at poetry. At first, she was able to stay at the Qingteng Study of Xu Wenchang, and later, she resided at Wushan in Hangzhou. She never held back, whether she was exchanging verses with luminaries everywhere, wielding her writing brush in front of her guests, or in the same room contending with a stag's tail (an implement held in discussions). 山陰王季重先生有八子，惟女玉映能讀父書，負才工詩. 初得徐文長青藤書屋居之，繼又寓武林之吳山，與四方名流相倡和，對客揮毫、同堂角塵所不吝也 (Ruan 2002, 40:471).

There is also the following description of Wang Duanshu by Chen Qinian 陳其年 (1626–1682):

> Wang Duanshu of Shanyin has a straightforward temperament, and she is especially strong at the study of history. Her father, Jizhong, would often caress her lovingly and say, 'I have eight sons who are not as cultivated [combined] as one daughter'. 山陰王端淑，意氣犖犖，尤長史學. 父季重常撫而愛憐之，曰，'身有八男，不易一女' (Ruan 2002, 40:471).

Wang Duanshu lived up to her father's wishes by having poetry correspondences with famous people everywhere, giving free rein to her poetic talents, producing many compositions, and even gaining the reputation of the best woman poet in the southeast (Zhou 2002, 10.622). Wang Duanshu married Ding Zhaosheng 丁肇聖 of Qiantang 錢塘, so she was the daughter-in-law of Ding Qianxue 丁乾學 (d. 1627). There is a record that states:

> Wang Duanshu's courtesy name was Yuying, and her alternative name was Yin-gran zi. She was the daughter of Mr. Suidong (Siren), and she was married to Ding Zhaosheng. She had broad learning. She was skilled in poetry and prose. She was good at calligraphy and painting; her specialty was painting flowers and grasses with sparse and forcefully beautiful brushstrokes. During the Shunzhi period, there was a desire to follow the precedent of Madame Cao Dagu, and Wang Duanshu was invited to the emperor's quarters to teach all the concubines and primary wives. She refused this. She died when she was more than eighty years old. As for her works, there is the *Yinhong ji* 吟紅集 (*Collection of Red Chants*) and the *Huazheng ji* 畫徵錄 (*Records of a Compilation of Painters*). 王端淑, 字玉暎, 號映然子. 遂東先生思任女也, 適錢塘丁肇聖. 博學工詩文, 善書畫, 長於花草, 疎落蒼秀. 順治中, 欲援曹大家故事, 延入禁中教諸妃主. 映然力辭之, 卒年八十餘, 著有《吟紅集》、《畫徵錄》(Feng 2001, 16:194).

Wang Duanshu was a talented woman who was proficient in poetry and prose as well as painting and calligraphy. With all that has been said, Wang Duanshu never renounced family life to become a bhikkhuni in her more than eighty years of living, and so, Yizhen En 一真恩 was not Wang Duanshu. I looked it up, and it seems that Yizhen En was Wang Duanshu's older sister, Wang Jingshu 王靜淑. Wang Jingshu was also known as Yizhen Daoren 一真道人, Yuyin 玉隱, Shi Jinglin 釋淨琳, and Yinchan zi 隱蟬子.[8] The *Mingyuan shihua* 名媛詩話 (*Poetry Notes of Eminent Ladies*) states:

> Wang Yuying (Jingshu) of Shaoxing. Her alternative name was Yinchan zi. She was the daughter of Wang Siren, who was an assistant commissioner (*qianshi* 僉事) of the former Ming dynasty. As for her writings, there is the *Qingliang Collection*. When her husband died, she preserved her chastity. Her 'Qiuri anju' 秋日菴居 ('Autumn Cloister Dwelling') says: 山陰王玉隱, 靜淑, 號隱禪子, 為前明僉事思任女, 著有清涼集. 夫亡, 清節自守.《秋日菴居》云：

I pass the late night in an empty room

空齋度深夜

Living as a hermit on a bed in the autumn.

高臥一床秋

Old, cold, colorless moss.

苔老寒無色

Clear rivulet; flowing fast in its shallows.

溪清淺欲流

Dust swept up with red leaves;

塵隨紅葉掃

A mind taken in by white clouds.

心付白雲收

Hearing the migrating wild geese in the desolate autumn winds,

蕭瑟聞征雁

Adds ten thousand bushels of sorrow.

添將萬斛愁

Her younger sister was Yuying 玉映 (Duanshu 端淑), whose alternative name was Yingran zi 映然子. She was the wife of the scholar Ding Zhaosheng. Her writings include the *Yuyingtang Collection* and *Yinhong Collection* (妹玉映, 端淑, 號映然子, 諸生丁肇聖室. 著有《玉映堂集》、《吟紅集》) (Shen 2002, p. 320).

In the *Wanqing yi shihui* 晚晴簃詩匯 (*Little Evening House Poetry Anthology*) is written:

Her courtesy name was Yuying 玉隱. Her alternative name was Yinchan zi 隱禪子. She was the eldest daughter of [Wang] Siren 思任 and wife of Chen Shurang 陳樹勷. She had the *Qingliang Collection* and the *Qingteng Studio Collection*. […] Yuyin was widowed when she was young. Her poetic ability was on par with her younger sister Yuying 玉映, but her reputation was overshadowed by Yuying. 字玉隱, 號隱禪子, 浙江山陰人. 思任長女, 陳樹勷室. 有清涼集、青藤書屋集 […]玉隱早寡, 詩才與妹玉映相伯仲, 名為玉映所掩(Shen 2002, p. 320).

Perhaps being bereaved because of her husband's death at a young age caused Wang Jingshu to renounce family life and become a bhikkhuni hermit, and this is why she was not as famous as her younger sister. We can know from this that Yizhen En of the Taiping Cloister in Wuhu was born to a multitalented and prestigious literati family.

Regarding number 9, the bhikkhuni Fajing Hao 法淨皓, there are more detailed records on her in the *Wudeng quanshu* than for other Chan bhikkhuni. For example:

A daughter of the Sun 孫 family of Danghu 當湖, her father was Master [Sun] Linwu 孫臨鵡 of the Board of Waterways 水部. Her mother dreamed that it was allowing a strange monk from the western regions stay for a night that led to her birth. In her childhood, she was peculiarly outstanding, and she liked to take after her father and read his documents. After she grew up, she assisted her father to go and manage the tariff administration of the port of Wuhu 蕪湖. Not long after marriage, she came to realize that the ways of the world were unyielding, and she requested to shave her head. Her father permitted this. She then went to get the support of Feiyin [Tong]rong 費隱[通]容 of Jingshan 徑山 to wear the robes and receive the full precepts. 當湖孫氏女, 父為水部臨鵡公, 母夢西域異僧投宿而生. 丱歲, 奇穎喜讀父書, 長則佐父出主蕪關稅政. 於歸未幾, 知世相堅, 乃請剃髮, 父許之, 遂投徑山費隱容披緇登具 (Jilun 1975–1989, 71:353b23–c2).

According to this, it is clear that Fajing Hao was born in the family of Sun Linwu, a Board of Waterways official in Danghu, part of the prefecture of Jiaxing. From a young age she 'liked to read her father's documents' and was able to assist his father in managing the tariff administration of the port of Wuhu. Her ability to assist her father in his official duties shows clearly that she was a capable reader and writer. The port of Wuhu was a busy hub of transportation and trade, as shown by the seven-character quatrain poem of the Yongzheng period by Li Chonghua 李重華 (jinshi: 1724), entitled 'Wuhu guan yebo' 蕪湖關夜泊 ('Mooring Overnight at the Port of Wuhu'):

Stranded at a meander; ten thousand miles of merchant ships

萬里商船滯一灣

Headwinds promote this more; and few turn back on their trips.

逆風更助上流慳

A pontoon bridge is sprinkled by rain all night,

浮橋竟夕潺湲雨

This barrier is the traveler's bane and plight (Li 2011, 9:190).

惱殺行人是此關

We can see from the relevant records of the *Fuyan Feiyin [Tong]rong chanshi jinian lu* 福嚴費隱容禪師紀年錄 (*Annalistic Records of the Chan Master Feiyin [Tong]rong of Fuyan*) that the bhikkhuni Fajing Hao was no exception to benefiting from the social capital of the monastic and layperson talents that converged under the flourishing dharma lineage of her guru, Feiyin Tongrong. For example, the entry for 1655 details Feiyin Tongrong's interactions between the two worlds of monastics and laypersons:

> When the master was sixty-three, there was a severely prolonged period of dry weather, and a special messenger of Xuedou Shiqi 雪竇石奇 (Feiyin Tongrong's dharma brother; 1594–1663) came to check in on him. The various officials of Qinchuan 琴川 (in Changshu 常熟 county) urged [Feiyin Tongrong to come], and he accepted this invitation. He first stayed overnight at the Pufu Temple 普福菴. As he was coming into the Weimo 維摩 [Monastery], on the twenty-fourth day of the second month, the path was jam-packed with monastics and laypersons there to greet him, and officials and literati came one after another to visit him and inquire into the Way. 師六十三歲, 大旱, 雪竇石奇和尚專使慰問. 琴川眾紳促, 師即應請, 師先宿山之普福菴; 二月二十四日入維摩, 緇素相迎, 道路盈塞. 邑之紳衿, 參訪問道, 絡繹無間 (Zifu 1987, 2:190).

In this convergence of eminent literati and monastics, there was no lack of bibliophiles and distinguished publishers. There were also literati who took part in publishing the Jiaxing Buddhist canon. For example, Feiyin Tongrong once gave a poem to Jiang Nangai 蔣南陔 (i.e., Jiang Fen 蔣棻; courtesy name Wanxian 畹仙; alternative name Nangai 南陔; 1605–1664). Jiang Fen was a native of Changshu who was in close contact with other literati from Changshu, such as Qian Qianyi 錢謙益 (1582–1664) and Mao Jin 毛晉 (1599–1659); he also participated in the later stages of the publication of the Jiaxing Buddhist canon,[9] such as assisting in the publication of Qian Qianyi's *Da Foding shoulengyan jing shu jiemeng chao* 大佛頂首楞嚴經疏解蒙鈔 (*An Ignorance-Removing Analysis of the Shoulengyan Scripture of the Greatest of the Great Buddhas*) for the Jiaxing canon. See Table 4.

**Table 4.** Sponsors of the *Da Foding shoulengyan jing shujie meng chao* 大佛頂首楞嚴經疏解蒙鈔 (*Explicatory Notes on the Da Foding Shoulengyan Jing Shu (Commentary on the Śūramgama-sūtra)*) (edition included in *J* vol. 18).

| | | |
|---|---|---|
| Cover, Part I | Textual reviewer: He Yun, worshiper of the Buddha from Yushan, at Baoen Monastery in Wulin, summer of the Wu-Xu year 戊戌夏佛弟子虞山何雲校勘於武林報恩院 | p. 119 |
| Cover, Part II | Sponsor of the distribution: Mao Fengbao (i.e., Mao Jin), worshipper of the Buddha from Yushan<br>佛弟子虞山毛鳳苞發願流通 | p. 127 |
| *Juan* 1.1 | Sponsor of the carving of printing boards: Xiao Bosheng, worshipper of the Buddha from Taihe<br>佛弟子泰和蕭伯升開板 | p. 139 |
| *Juan* 1.2 | Sponsors of the carving of printing boards: The masters of Lingyan Hongchu, [2 lacunae], Cuitang, Sengjian, Wuying, Shengchu, Dayuan, and Yuehan<br>靈嚴和尚弘儲<br>□□ 翠堂 僧鑒 物英 聖初 大圓 月函開板 | p. 147 |
| *Juan* 1.3 | Sponsors: [lacuna] Zhang [5 lacunae] *yi* (society)<br>□張□□ □□□邑助緣 * | p. 157 |
| *Juan* 2.1 | Sponsors: Zhang Youyu, Wang Shimin, Wu Weiye, Chen Hu, Xu Bo, Shen Minglun, Zheng Fujiao, Wang Tingbi, Wang Ting, Wang Gui, Zhou Yunxiang, Gu Mei, Xu Kairen, Qian Gu, Ye Guohua, Huang Kan, Li Xunzhi, Zheng Qinyu, Lu Xianbi<br>張有譽 王時敏 吳偉業 陳湖 徐波 沈明掄<br>鄭敷教 王廷璧 王挺 王揆 周雲驤 顧湄<br>徐開任 錢畻 葉國華 黃侃 李遜之 鄭欽諭 陸獻陛助緣 | p. 166 |



**Table 4.** *Cont.*

| | | |
|---|---|---|
| *Juan* 2.2 | Sponsors: Sun Chaorang, [3 lacunae], Zhao Shichun, Shao Deng, Zhao Yanxian, Chen Shi, Chen Suhuang, Sang Wo, Xue Weiyan, Wang Ruxuan<br>孫朝讓 □□□ 趙士春 邵燈 趙延先 陳式 陳溯潢 桑沃 薛維嚴 王入玄助緣 | p. 177 |
| *Juan* 2.3 | Sponsors: Zhou Min, Shen Rulan, Mao Shiya, Xi Qiyuan<br>周敏 沈汝蘭 毛詩雅 席啟圓助緣 | p. 181 |
| *Juan* 3.1 | Sponsors: Jiang Fen, Gui Qixian, Yan Yang, Qu Xuanxi, Chen Huangtu, Xu Qi, Dai Mi, Jun Xun<br>蔣棻 歸起先 嚴恙 瞿玄錫 陳煌圖 許琪<br>戴泌 浚洵助緣 | p. 191 |
| *Juan* 3.2 | Sponsors: Yan Shi, Lu Tingbao, Lu Tingfu, Wang Fenglai, Xu Wenwei, Lu Wenhuan, Lu Lu/He, Zeng Zhaojia<br>嚴栻 陸廷保 陸廷福 王奉來 徐文蔚 陸文煥<br>陸輅 曾肇甲助緣 | p. 200 |
| *Juan* 4.1 | Sponsors: Gui Hong, Mao Yi<br>歸泓 毛宸助緣 | p. 215 |
| *Juan* 4.2 | Sponsors: Qian Zushou, Qian Chaoding, Qian Zushou, Qian Zuxing, Qian Longti, Qian Guofu, Qian Lucan, Qian Chaonai, Qian Zanxian, Qian Qianguang, Qian Chenji, Qian Sunyan, Qian Yijia, Qian Wanxuan, Qian Zonglong, Qian Minzhong, Qian Sishan, Qian Qianheng, Qian Qianji, Qian Qianxiao, Qian Wanghuan, Qian Sunbao<br>錢祖壽 錢朝鼎 錢祖授 錢祖行 錢龍惕<br>錢國輔 錢陸燦 錢朝鼐 錢讚先 錢謙光<br>錢臣績 錢孫燕 錢裔嘉 錢萬選 錢宗龍<br>錢敏忠 錢思山 錢謙亨 錢謙吉 錢謙孝<br>錢王桓 錢孫保助緣 | p. 226 |
| *Juan* 5.1 | Sponsors: Sun Yongzuo, Lu Yidian, Wang Junchen, Wang Qingchen, Yan Xiong, Wu Peichang, Wu Longxi, Chen Hezheng, Xu Depan, Xu Dezhen<br>孫永祚 陸貽典 王俊臣 王清臣 嚴熊 吳培昌<br>吳龍錫 陳鶴徵 許德璠 許德珍助緣 | p. 231 |
| *Juan* 5.2 | Sponsor of the carving of printing boards: Xiao Mengfang, worshipper of the Buddha from Taihe<br>佛弟子泰和蕭孟昉開板 | p. 242 |
| *Juan* 6.1 | Sponsor of the carving of printing boards: Xiao Mengfang, worshipper of the Buddha from Taihe<br>佛弟子泰和蕭孟昉開板 | p. 250 |
| *Juan* 6.2 | Sponsors: Wang Yueyu, Wang Feng, Li Lin<br>王曰俞 王澧 李臨助緣 | p. 259 |
| *Juan* 6.3 | Sponsors: Sun Maoshu, Zhou Anren, Zhou Changsheng, Gu Maolun, Qi Zuogan, Zhu Heling, Wen Bing, Huang Tingbiao, Hu Bashui, Guan Yinglü, Su Minggao<br>孫茂叔 周安仁 周長生 顧茂倫 戚左干<br>朱鶴齡 文秉 黃廷表 胡八水 管應律 蘇鳴皋助緣 | p. 263 |
| *Juan* 10.3 | Sponsor of the carving of printing boards: Xiao Mengfang, worshipper of the Buddha from Taihe<br>佛弟子泰和蕭孟昉開板 | p. 345 |
| The *Da Foding shoulengyan jing shujie meng chao* "foding wulu"<br>《大佛頂首楞嚴經疏解蒙鈔》"佛頂五錄",<br>*Juan* 2, 4, 5, 6, 7, 8 | Sponsor of the carving of printing boards: Xiao Mengfang, worshipper of the Buddha from Taihe<br>佛弟子泰和蕭孟昉開板 | pp. 355, 380, 392, 405, 411, 417 |

\* One □ in this table represents one Chinese character that is illegible or missing in the original text.

Feiyin Tongrong also presented a verse to another literati from Changshu he had contact with named Sun Xiaoruo 孫孝若. Sun Xiaoruo served an as official in the Court of Judicial Review, and he was an important patron of Buddhism. Sun Xiaoruo was also a book collector, and he was in close contact with Qian Qianyi. In the *Cangshu jishi shi* 藏書紀事詩 (*Narrative Poems on Book Collecting*) is written:

*Airi jinglu cangshu zhi* 愛日精廬藏書志 (*Book Collecting Records of Seize the Day Studio*) *Li Shangyin Collection*, in three fascicles. The postscript by Chen Hong 陳鴻 (1780–1833) states, 'In the first month of the *bingxu* year, I borrowed Sun Xiaoruo family's Northern Song edition to collate'. Noted by [Ye] Changzhi [葉]昌熾 (1849–1917): Xiaoruo was Qian Lüzhi's 錢履之 son-in-law. In the *Huaijiu Collection* 懷舊集, there is a poem by Lüzhi entitled 'Jiuyue qiri Xiaoruo xu yizhuo lingshu Guangfu xinge cheng Guangfu ding jiuri denggao zhiqi yaotong Cangxi gong bei shanxing jishi shi' 九月七日孝若婿移酌令叔光甫新閣承光甫訂九日登高至期邀同己蒼夕公輩山行紀事詩 ('A Narrative Poem of a Mountain Hike, Son-In-Law Xiaoruo Went Drinking at Your Uncle Guangfu's New Pavilion', 'Got Guangfu to Agree to Plans for the Double Ninth Climbing High Day', and 'When the Day Came Cang Xigong and Others Were Also Invited'). This poem states:

《愛日精廬藏書志》：'《李商隱集》三卷，陳鴻跋曰：丙戌正月，借孫孝若家北宋版校'. 昌熾案：孝若為錢履之之壻，《懷舊集》有履之《九月七日孝若壻移酌令叔光甫新閣承光甫訂九日登高至期邀同己蒼夕公輩山行紀事詩》，詩云：

The year was *jiashen*, the month was nine

　　　　　　　　　　　　　　　　　　　　　　　歲惟甲申月在九

Nephew Sun, soup cakes, pouring happy wine.

　　　　　　　　　　　　　　　　　　　　　　　孫甥湯餅酌喜酒

Since he called Guangfu the uncle of Xiaoruo, then clearly, Xiaoruo must be the son of [Sun] Chaosu [孫]朝肅 (1584–1635). I think his Northern Song edition of the *Li Shangyin Collection* was a book passed down in his family as a copy his father got from Mr. Hua 華氏. Xigong 夕公 was the courtesy name of Qian Longti 錢龍惕 (1609–?). Longti has poems entitled 'Xiaoruo suocang Chanyue dashi hua shiliu luohan ge' 孝若所藏禪月大師畫十六羅漢歌 ('A Song on the Painting of the Sixteen Arhats by Great Master Chanyue, in the Collection of Xiaoruo') and 'Tong Xiaoruo song Silong gui Wumen' 同孝若送士龍歸吳門 ('Together with Xiaoruo Seeing Off Shilong on His Return to Wumen'). Feng Zhongshu 馮仲舒 wrote a poem, 'Sun Xiaoruo Fangguang ju zhaokan Bamian chongtai mudan' 孫孝若方廣居招看八面重臺牡丹 ('Sun Xiaoruo's Expansive Residence, Invited to See Eight-Faced Double-Terraced Peonies'). Also, according to the *Tianlu linlang xubian* 天祿琳瑯續編 (*Continued Catalogue of the Imperial Palace Library*), a Guangzong period woodblock edition of the *Zuantu huzhu Shangshu* 纂圖互注尚書 (*Mutually Annotated and Illustrated Classic of Documents*) has the 'Prince of Kui 夔王 family seal of Sun Fan 孫藩 of Yushan 虞山'. 既稱光甫為孝若之叔，則孝若為朝肅子可知. 其所藏北宋本《李集》，即功父所得華氏之本，蓋其家楹書也. 夕公，錢龍惕之字，龍惕有《孝若所藏禪月大師畫十六羅漢歌》一首，又有《同孝若送士龍歸吳門》詩. 馮仲舒有《孫孝若方廣居招看八面重臺牡丹》詩. 又按：《天祿琳瑯續編》；'《纂圖互注尚書》，宋光宗時刻本，有'虞山孫藩夔王氏之印' (Ye 1991, 3:326–27).

It is clear from this that Sun Xiaoruo's family collected precious editions of books and paintings. Xiaoruo was a renowned collector, the eldest son of Sun Chaosu, and the son-in-law of Qian Lüzhi. Qian Longti was a nephew of Qian Qianyi, who wrote poems about Sun Xiaoruo. Sun Xiaoruo family had a longstanding tradition of erudition. As Wu Weiye 吳偉業 (1609–1672) said, 'The elegant radiance of his genius, and the inheritance of his family's learning, enables him to bring together ancient and modern, penetrate the classics and the histories, and move between classical prose and poetry' (其天才之所軼發，家學之所纘承，足以囊括古今，貫穿經史，出入古文詩歌之間).[10] Xiaoruo was a man of abundant talent. In his 'Sun mu Guo ruren shouxu' 孫母郭孺人壽序 (*On Wishing Long Life to Sun's Mother Lady Guo*), Wu Weiye mentions his intimate relationship

with Sun Xiaoruo's family, starting with his admiration for Sun Xiaoruo's grandfather Sun Ziqiao 孫子喬. Wu Weiye also passed the imperial examinations like Sun Xiaoruo's father, Chaosu (Gongfu 恭甫), in the same year as had Sun Xiaoruo's uncle Chaorang 朝讓 (Guangfu 光甫; 1593–1682), and Sun Xiaoruo's paternal half-brother Xiaowei 孝維 was his student. Therefore, on the fiftieth birthday of Sun Xiaoruo's mother, Wu Weiye offered his congratulations of long life in succession with Qian Qianyi and Sun Guanfu as follows:

> Xiaoruo is nimble and refined in his speech and bearing and can put a piece of writing together. He is skilled in rhetoric and brushwork. He exchanges mutual encouragement with the greatest of the world by his honor and affable air. By this, I know that his mother is virtuous and knows how to teach…Xiaoruo is a gentleman who knows many things. He is an expert at collecting. As for what his forebears have left to him, he has an outstanding ability to care for these inherited relics. By this I know that his mother is respectful and knows how to protect. […] My student Xiaowei is Xiaoruo's younger brother of a different mother. […] As for the other young men of the families of Gongfu and Guangfu, these sprouting orchids are sturdy as jade; however, I would choose Xiaoruo as the superior one to work as an illustrious official. […] The good fortune of the Sun family is like the sun rising over the river; it truly does not end. And if we trace back the origin of this second generation, it truly begins with Xiaoruo's mother. […] The reason why I dare to follow Qian Qianyi, and Guangfu, in presenting my humble words is that I have a close relationship with Xiaoruo and I know everything about his mother. Therefore, I do not hesitate to praise and pray for her. May she live many more years. 孝若姿神吐納, 警速風流, 好屬文, 工詞翰, 交天下賢豪長者, 以名節氣誼相砥勵. 吾以知孺人之賢而能教也. ……. 孝若博物君子, 雅擅收藏, 而於先世所遺, 尤能護持手澤, 吾以知孺人之敬而能守也. ……余門人孝維, 為孝若之異母弟, ……蘭芽玉茁, 而孝若掇上第, 就顯官, ……. 蓋孫氏之福澤, 如日升川至, 正未有艾, 而遡其再世發祥, 實啟自孺人, ……余所以隨牧齋、光甫兩公之後, 敢具不腆之詞進者, 實以交於孝若者深, 知於孺人者悉, 故不憚靦縷以致其頌且禱也. 是為壽 (Wu 1975, pp. 1023–24).

While Sun Xiaoruo interacted with Feiyin Tongrong, Sun's wife also was among a number of the women members of prominent literati families who had exchanges in which they asked to be taught Buddhism by the aforementioned Chan bhikkhuni Jizong Che 季總徹 (b. 1606) (Jizong 1987; Su 2016, p. 55). These exchanges can be seen in sections of Jizong Che's recorded sayings that include 'Shi Qian Muzhai furen' 示錢牧齋夫人 ('Presenting to Madame Qian Muzhai'), 'Shi Qian Fuxian furen' 示錢復先夫人 ('Presenting to Madame Qian Fuxian'), 'Shi Sun Xiaoruo furen' 示孫孝若夫人 ('Presenting to Madame Sun Xiaoruo'), and 'Yu Gao furen' 與高夫人 ('Presenting to Madame Gao'). Qian Muzhai is Qian Qianyi, and his wife, as mentioned earlier, also had exchanges with the bhikkhuni Yuanjian Xingxuan.

Feiyin Tongrong's exchanges with the Buddhist monastic community included letters or visits with his dharma brother Muchen Daomin 木陳道忞 (1596–1674), his disciple Yinyuan Longqi 隱元隆琦 (1592–1673), his dharma grandnephew Huotang Zhengyan 豁堂正嵒 (1597–1670), and Fachuang Xingzhi 法幢行幟 (1593–1667).[11] Feiyin Tongrong said in the fall of 1655, 'The bhikkhuni Fajing Hao of Pinghu came to visit. Hao has always taken refuge under me. There is an affinity between us, and she has been coming to ask for my teachings for a long time. I bestowed upon her a *ruyi* as a token of her confirmation' (平湖比丘尼法淨皓來省觀, 皓素皈依座下, 有機緣相契, 往來問道甚久, 付如意表信) (Zifu 1987, 2:190).

From this, we can see that Feiyin Tongrong approved of the Chan bhikkhuni Fajing Hao and that he gave her a *ruyi* as a seal of approval. At the end of Feiyin Tongrong's recorded sayings are lists of his monastic and layperson disciples who participated in the publication of his recorded sayings. First, there is a list of his monastic disciples as follows: Longqi 隆琦, Xingji 行璣, Xingmi 行彌, Xingyuan 行元, Xingmi 行密, Xingding 行定, Xingjian 行鑑, Xingcheng 行成, Xingsheng 行盛, Xingsi 行巳, Xingzong 行宗, Xing-

min 行敏, Xingzhen 行真, Jitan 寂坦, Jifu 寂枈, Xingzhou 行舟, Xingguang 行廣, Guiquan 規權, Daquan 大全, Xingguan 行觀, Xinggu 行古, Xinghong 行宏, Xingduan 行端, Xingxue 行雪, Xingfa 行法, Xingli 行立, Xingqian 行謙, Xingran 行然, Xingqian 行潛, Xinglang 行朗, Xingjing 行淨, Xingming 行明, Xingzhen 行臻, Xingji 行濟, Xingren 行仁, Xingzhong 行中, Xingjing 行敬, Xinggao 行高, Xingjing 行鏡, Xingzhan 行湛, Xingchen 行琛, Xingbei 行備, Xingsheng 行省, Xingling 行淩, Jitai 寂泰, Yuanshun 元順, Jiqian 寂乾, Zhanying 湛瑩, Jiding 寂定, Xingzhi 行志, Xingyin 行蔭, Xingfu 行珗, Zhijing 智經, Xingjian 行澗, Xinghui 行會, Ruxin 如信, and Xingjun 行俊. Then there is a list of his layperson disciples Wang Gu 王谷, Yan Dacan 嚴大參, Li Zhongzi 李中梓, Yan Xingda 嚴行達, Xu Changzhi 徐昌治, and Dong Xingzheng 董行證. Only at the end, after all these other disciples of Feiyin Tongrong is the last name that appears, the bhikkhuni disciple Fajing Hao, who is listed under the name Xinghao 行皓. Whether her name being listed after the male monastics and laymen indicates a ranking by gender in Chan Buddhism is a question for another article. However, from the fact that so many monastics and lay disciples of Feiyin Tongrong joined together to publish his recorded sayings, we can infer that the inclusion of these records in the canon is related to this joint effort of his many disciples, and the promotion and sponsorship of laypersons also seem to be an undeniably important source of support for the publication for these records. For example, Yan Dacan 嚴大參 (1590–1671; Duoli Daoren 軲道人), who participated in the publication of these records, also helped to promote the publication of the recorded sayings of Jizong Che that were included in the Jiaxing canon. We can also frequently see that Yan Dacan supported other publications in the Jiaxing canon in the early Qing period, including a commentary of the *Huayan jing* 華嚴經. Such a rich network of supporters for Feiyin Tongrong must have influenced the bhikkhuni Fajing Xinghao's attention to the compilation and publication of his works. A colophon at the end of these recorded sayings reads:

> Madam Cheng 程氏 of Caomen 曹門 in the district of Shimen 石門, whose dharma name is Xingning 行寧, and the bhikkhuni Xinghao 行皓 of Pinghu 平湖, who was a dharma heir disciple, jointly funded the engraving of the annalistic records of Feiyin Tongrong to circulate widely for the benefit of future students. It is our sincere wish that the dharma will continue to flourish and that the mountains and rivers will be forever peaceful. 石門縣曹門太夫人程氏, 法名行寧. 同平湖縣嗣法弟子尼行皓, 共奉貲敬刻福嚴費老和尚紀年錄壹冊, 共全錄流通, 以惠後學. 伏願法道遐昌, 山河永靖者(Fenyin Tongrong 1987, 14:193a).

Feiyin Tongrong's laywoman disciple Xingning 行寧 and her woman dharma heir Fajing Xinghao jointly funded the publication of the *Fuyan Feiyin Rong chanshi jinian lu* so that it could circulate widely. Their dedication to the publication of these records and Feiyin Tongrong's recorded sayings shows their respect for their teacher and the value they attached to publishing his works.

Chan bhikkhuni number 10 is Zhaoqing Guanghao 照清光皓, number 11 is Yinyue Xinglin 印月行霖, and number 19 is Renfeng Jiyin 仁風濟印. Numbers 10 and 19 both came from the families of prominent officials. Zhaoqing Guanghao was the 'granddaughter of Master Zhang Wenzhong 張文忠, the grand secretary (*xiangguo* 相國) from Yongjia 永嘉', meaning that she was a granddaughter of Zhang Cong 張璁 (1475–1539), who was a grand secretary in the Jiaqing period of the Ming dynasty. Renfeng Jiying was a great-granddaughter of Gu Dingchen 顧鼎臣 (1473–1540), a grand secretary from Kunshan 崑山, also known as Master Gu Wenkang 顧文康公. Yinyue Xinglin was from a famous literati family and a niece of Huang Taichong 黃太沖 (i.e., Huang Zongxi 黃宗羲; 1610–1695) of Yaojiang 姚江. It would not have been difficult for these three bhikkhuni to develop writing abilities in poetry and prose. However, little more can be known about the families or secular names of these three women. For example, as mentioned before, the bhikkhuni Zhaoqing Guanghao, who was the granddaughter Zhang Cong, was thirteen when her father tragically died. She thereupon began to uphold a Buddhist-style vegetarian diet and vowed to never speak again. According to the historic record, Zhang Cong had three sons, and the fate of his second son, Zhang Xunye 張遜業 (1525–1560), was most consistent with

Zhaoqing Guanghao's childhood experience. According to the biography of Shen Lian
沈鍊 (1507–1557) in the *Mingshi* 明史 (*History of the Ming*), Shen Lian and his good friend
Zhang Xunye met with misfortune upon going against and exposing the many misdeeds
of Yan Song 嚴嵩 (1480–1567) and his son (Zhang 1974, 209:5533–35). Therefore, if the
bhikkhuni Zhaoqing Guanghao was the daughter of Zhang Xunye, she would have been
thirteen years old when her father died in 1560, and so, she would have been born in 1548.
However, if this was her birth year, she would have been born eighty years before her
teacher, Tianmu Chaozhi 天目超智 (b. 1626), and that seems extremely unreasonable.

Yinyue Xinglin was a niece of Huang Zongxi. According to my research, her father
was Huang Jindong 黃金棟 and her mother was surnamed Pan 潘.[12] At the age of sixteen,
she married into the Xie 謝 family of Dongshan 東山, and three years later, she put on her
Buddhist robes and took the monastics precepts at Li'an 理安 in Hangzhou (Shi 1983, 5:85).
She authored the *Fulong Yinyue chanshi yulu* 伏龍印月禪師語錄 (*The Recorded Sayings of
Chan Master Yinyue of the Fulong Cloister*).[13] However, there is still insufficient historical
evidence to confirm whether Yinyue Xinglin was in fact the daughter of Huang Jindong
and whether Huang Jindong was the cousin of Huang Zongxi. Further research is needed
to confirm this.

Number 19, Renfeng Jiyin 仁風濟印, had a fourth-generation descendant of Gu
Dingchen 顧鼎臣 (1473–1540) as his great-granddaughter. By this reasoning, she must be a
sibling of Gu Xianzheng 顧咸正 (Master Gu Wenkang of Kunshan 崑山顧文康公; d. 1647),
Gu Xianjian 顧咸建 (d. 1645), and Gu Xianshou 顧咸受. At a time of vulnerability, when
the soldiers of the Manchu Qing came south, the Gu Xianzheng brothers fought and died
for their country one after another, while refusing to surrender. It is recorded in the section
of 'Gu Xianzheng zuo Wu Shengzhao shi si' 顧咸正坐吳勝兆事死 ('Gu Xianzheng Died in
Wu Shengzhao's Enterprise') of the *Mingji nanlüe* 明季南略 (*Southern Campaigns during the
Ming Period*):

> Gu Xianzheng's courtesy name was Duanmu 端木, and his alternative name was
> Ji'an 皈庵. He was from Kunshan 崑山. He was the great-grandson of Master
> Wenkang 文康公 (i.e., Gu Dingchen). He was the elder brother of [Gu] Xianjian
> [顧]咸建 (?–1645). He became a provincial graduate (*juren* 舉人) in the *guiyou* year
> of the Chongzhen reign (1633), and in the thirteenth year (a *gengchen* year) (1640),
> he was placed on the supplementary list to be appointed as a judge of the pre-
> fecture of Yan'an 延安府. […] When Wu Sangui's 吳三桂 (1612–1678) soldiers en-
> tered Qin 秦 (i.e., Shaanxi 陝西), many rose to respond [in support]. The people of
> Hancheng 韓城 nominated Xianzheng to be their leader, and they beheaded the
> illegitimate commander Wang Yechang 王業昌. He soon knew that these were
> soldiers of the Great Qing, and so, he went into the mountains. 顧咸正字端木,
> 號皈庵; 崑山人, 文康公之曾孫、咸建兄也. 崇禎癸酉舉人. 十三年庚辰, 以副榜除
> 延安府推官. ……吳三桂兵入秦, 人多應之. 韓城人推咸正為主, 斬偽令王業昌. 已
> 而知為大清兵, 遂入山中.

> The next year he returned south with all of his hair intact. With the defeat of the
> enterprise of Wu Shengzhao 吳勝兆 (?–1647) and Chen Zilong 陳子龍 (1608–1647)
> in Yunjian 雲間, the names of the members of their faction were recorded, begin-
> ning with Xianzheng. Thereupon, Xianzheng died along with forty persons of
> this same enterprise. Xianzheng's sons [Gu] Tiankui [顧]天逵 (1618–1647), a trib-
> ute student (*gongsheng* 貢生) with the courtesy name Dahong 仲熊, and [Gu]
> Tianlin [顧]天遴 (1621–1647), a licentiate (*zhusheng* 諸生) with the courtesy name
> Zhongxiong 仲熊, also died for having hidden Chen Zilong. […] At that time, the
> local magistrates were overpowered into submission everywhere the soldiers of
> the Great Qing passed through, and few officials died for their cities. Only Gu
> Xianzheng's younger brother Xianjian, his courtesy name Haishi 海石 and his
> alternative name Ruxin 如心, was a presented scholar of the *guiwei* year of the
> Chongzhen reign (1643), and he was appointed district magistrate (*zhixian* 知縣)

of Qiantang—he was killed for being unyielding while being captured for burning the registers. [...] Xianzheng's youngest brother, Xianshou 咸受 (?–1645), was a provincial graduate of the *jiazi* year of the Tianqi reign (1624), his city was defeated, and he also died. Xianzheng was survived only by his grandson Jingu 晉穀 (otherwise unknown), who was exonerated for being only five years old. The Dahong brothers themselves said, 'We have received the grace of the dynasty for generations; although we are literati, righteous does not allow live in dishonor'. Thus, five fathers and sons of one family together died in service of their dynasty. The literati of the Suzhou area were all saddened by this. 又明年, 以全髮歸南. 會雲間吳勝兆、陳子龍事敗, 錄其黨姓名, 首及咸正; 乃與同事四十餘人並死. 子天遴字大鴻, 貢生; 天遴字仲熊, 諸生. 皆以藏子龍故, 亦死. ……是時大清兵所過州縣, 從風而靡, 長吏罕有殉城者. 獨公弟咸建字海石, 號如心; 崇禎癸未進士, 除錢塘知縣. 以焚冊故, 被擒不屈, 殺之. ……咸正季弟咸受, 天啟甲子舉人. 城破, 亦死. 僅存一孫晉穀, 年五歲, 得免. 大鴻兄弟自謂 '世受國恩; 雖書生, 義不苟活'; 故一門父子、五人同死國事. 吳中人士, 莫不悲之 (Ji 1963, 9:278–79).

Gu Xianzheng, his two brothers, and his two sons all died as Ming loyalists during the turmoil of the Ming–Qing transition. As Renfeng Jiyin was a member of this Gu family that suffered this great misfortune, we can imagine that she probably made her decision to shave her head and become a bhikkhuni of the Chan school as a way to move forward with her life during a period of troubled times.

As for the relevant life events of number 20, the Chan bhikkhuni Lingxi Rong, we only know about an interaction between Linxi Rong and her teacher Jubo Jiheng, with no further description of her background (Jilun 1975–1989, 103:621c14–23). However, we can obtain some relevant information from the records of her dharma brother Venerable Shangsi 上思 (1630–1688), who was also known as Chan Master Xuewu Yushan 雪悟雨山. We read in the section 'Lingxi chanshi wuzhi xu' 靈璽禪師五秩序 ('On Wishing Continued Long Life to Chan Master Lingxi on Her Fiftieth Birthday') of Shangsi's *Yushan heshang yulu* 雨山和尚語錄 (*Recorded Sayings of Venerable Yushan*):

I read the Chan histories, and in the Song, there was such a person of Moshan 末山 who incarnated as a bhikkhuni while possessing the functioning of an outstanding source teacher. [...] From the Song and the Yuan and the Ming and up to [the present] sagacious dynasty, over a period of more than five hundred years, I have read the records of Chan meticulously, and there are none who could succeed Moshan's excellent doings. And now, Master Lingxi Rong manifests these! The master was born into an illustrious family, and from her infancy, she had the resolve to transcend the defiled world. She viewed the wealth of the world and its many splendid things as like an illusory bubble or a flash of lightning. If her heart were not as strong as a *kalpa* stone, and her resolve not as staunch as the autumn frost, how could she attain this? Still, it is not that the master was not without help, and so, she was able to prosper like this. I think she is of the same mind and conviction as Master Tianjing Che 天鏡徹, and perhaps, their relationship is comparable to that between Vasubandhu and Asaṅga. I remember that in the winter of the *xinchou* year (1661) when the two masters came to Tianning to receive the precepts, they at once had a consultation and asked for instructions of the ultimate one move to the highest enlightenment. The late guru instructed them, as appropriate, and they both attained a treasure as if excavating what was buried [within themselves]. The late guru soon after wrote at his gate the four characters 'Moshan fengjing' 末山風境 ('The Scenery of Moshan') with the hope that the two masters would understand, and it was not long after this that he died. He knew that they would be capable of protecting the great teachings in the future, and so, he gave his robes to them. The two masters were all the more deeply moved by this gesture, and from then on, they encouraged

each other with the hope that they could penetrate the principle of the dharma. Now, there are several verses written by each of them, many of which are marvelous in their clarity. Is not this evidence of their endowment of a firm and stable foundation? Moreover, they have a kind of mental strength in their life-long dedication to the propagation and protection of Buddhism, which is so unparalleled by Moshan and her like that I cannot even speak of them together on the same day. 予往閱禪史, 至宋時有末山其人者, 而以尼比丘身, 具宗師峻拔之用. ……自宋而元而明以及聖朝, 歷五百餘年, 細閱禪典, 無可以繼其芳躅者, 乃今于靈璽融公見之矣. 公生本華冑, 襁褓時即具出塵之志. 視世利紛華如幻泡電影. 非心堅劫石, 志烈秋霜, 曷克臻此? 然公非無所助, 遂能發祥爾爾. 蓋與天鏡徹公同德同心, 殆世所謂天親無著也. 憶辛丑冬. 二公以秉戒來天寧, 即以向上一著咨扣. 先師隨宜開發, 俱獲神珠, 如發伏藏. 既而以'末山風境'四字署其門, 則其屬望二公可知. 未幾, 先師往化, 知其將來堪任大法, 即以己衣分授. 二公益感激, 由是互相策發, 期以徹法源底. 今所著偈頌各若干首, 多圓妙昭徹, 豈非沖厚鎮奠之資之一驗歟? 況平生弘護法門, 一種心力, 又非末山輩所可同日語也 (Shangsi 1987, 19:608).

The bhikkhuni Lingxi Rong's dharma brother Yushan was born into the Yu 于 family of Taizhou 泰州 in the prefecture of Yangzhou 揚州, and he resided at the Tianning Monastery of Yangzhou. The 'Yangzhou Tianning Yushan Si heshang taming' 揚州天寧雨山思和尚塔銘 (Inscription on the Stupa of Venerable Yushan Si at Tianning Monastery in Yangzhou') records that in 1682, both the monastic and lay communities of the Weiyang 維揚 region of Yangzhou strongly advocated for Yushan to manage the Tianning Monastery:

The patrons of Weiyang, and all the elders of the monastery, discussed with each other how it was impossible to revitalize the Tianning Monastery without Yushan, and so, they wrote a letter and appointed a messenger to earnestly invite him from afar. Yushan thought that it was unacceptable for Chan to be left empty for a long time, and he also felt troubled to go on ignoring everyone's feelings forever, and so, he immediately set out [to return to the Tianning Monastery]. […] When Jibo Jiheng of the Tianning Monastery passed away into nirvana, his successors were not yet refined in selecting out the pure milk [of the dharma] and the old ways were nearly toppled. As soon as Yushan arrived, unanimous praise blew across the Yanghuai region. He revived and revitalized the monastery without any extra effort, and he took care of everything neglected inside and outside of the gates of the monastery. 維揚檀越諸山耆舊, 僉議天寧一席, 非公不能振起. 乃削牘命使遠致懇誠. 公念祖席不可久虛, 輿情亦難終負, 遂幡然而起. ……天寧恒和尚唱滅, 繼席者擇乳未精, 典型幾覆. 公一至而翕然稱善, 風動江淮, 摧闢振興, 不資餘力. 山門內外, 百廢具舉 (Shangsi 1987, 19:615).

When Emperor Kangxi came to Yangzhou on his southern tour of 1684, he 'first visited Tianning Monastery' 首幸天寧 (Shangsi 1987, 19:615). The Tianning Monastery frequently served as a temporary imperial residence when Qing emperors came south, and it was one of the great and celebrated Buddhist monasteries of the Jiangnan region. The Chan monks who were able to stay at this monastery must have been the eminent who were famous everywhere:

Yangzhou was a key throughfare between the north and south to which all vessels or carriages would arrive. Every day, when Yushan had not washed up yet [at daybreak], the outside of his door was already brimming with footsteps [of visitors to the Tianning Monastery]. Yushan always took the preservation of the dharma as his personal responsibility, and he was never negligent. 揚當南北要衝, 舟車畢至. 日未盥漱, 戶履已盈. 公素以荷法為己任, 始終不少懈息 (Shangsi 1987, 19:615).

Clearly, the travel hub of Yangzhou and the easy accessibility of the Tianning Monastery allowed it to be a gathering place of the social and economic capital, which were important prerequisites to becoming famous. Therefore, the bhikkhuni Lingxi Rong, who

resided in the Jiangdu 江都 district of the prefecture of Yangzhou, had access to many local and dharma connections. She benefited from a supportive relationship with her dharma brother Yushan, and perhaps, she was even able to benefit from his relatively rich interpersonal resources to establish her comparatively advantageous position in the Chan lineage.

Number 25, the Chan bhikkhuni Jingnuo Yue 靜諾越, was a dharma heir of the Chan bhikkhuni Weiji Zhi 惟極致 (d. 1672) of the Linji denomination. Jingnuo and her guru, Wei Jizhi, were both women poets who were skilled with a brush and ink. Weiji Zhi was a woman from an eminent family of Yaojiang 姚江, and Jingnuo was also a boudoir flower of the illustrious Lin 林 family of Renhe 仁和 (Shen 2002, 12:693). We read in the *Liangzhe youxuan xulu* 兩浙輶軒續録 (*Continued Records of Zhejiang Poetry by an Imperial Emissary*):

> Jingnuo's alternative name was Zixian Daoren 自閒道人. She was from Renhe 仁和 and was a bhikkhuni at the Xiongsheng Convent 雄聖庵尼. She wrote *Draft Poems of Shoulder-Resting Cottage*. 靜諾號自閒道人, 仁和人, 雄聖庵尼, 著《息肩廬詩草》(Ruan 2002, 54:246–47).

The work then quotes three of Jingnuo's poems entitled 'Suimu zuo' 歲暮作 ('Written at the Year's End'), 'Zhoubo Gushu' 舟泊姑蘇 ('Boat Mooring in Gushu'), and 'Hezhu chuxia' 河渚初夏 ('Early Summer on a Riverbank').

Number 37, Lianhua Kedu 蓮花可度, and number 38, Mingxin Foyin 明心佛音 (d. 1674), were both Chan bhikkhuni dharma heirs of Jie'an Wujin 介菴悟進 (1612–1673), making them related as dharma sisters. Both suffered the loss of their families as children, and this motivated both of them to be determined to transcend the defiled world. However, records indicate that Lianhua Kedu was the youngest child of the Tian 田 family of Huai'an 淮安 and that her father was an official. Her father lost his life for a crime of misconduct when she was seven years old. Perhaps this led to her family's decline. Mingxin Foyin was a daughter of the Ye 葉 family of Puzhen 濮鎮 in Zuili 檇李. She lost her father, Jianzai 健在, at a young age, but her family situation is still unclear. We can see that she had a hardworking and frugal side because she began 'several years of ascetic practice' (苦行數載) after becoming a Buddhist monastic and on one occasion 'mended her woolen robe at Luo Guangwen's house' (在羅廣文家補毳) (Jilun 1975–1989, 81:441b12–20).

As seen in the aforementioned examples, when these literati women of illustrious families became Chan bhikkhuni, it was not only their own personal cultivation of their talents and accomplishments but also the family and regional connections that were connected to their backgrounds that played a role in their cultivation as Buddhist monastics. Their change in identity did not seem to interrupt the support they had, and sometimes, it seems that they were able to expand their networks of support to both the layperson and monastic communities after becoming bhikkhuni.

## 5. The Support Systems Underlying the Achievements of Chan Bhikkhuni

Of the thirty-eight Chan bhikkhuni listed in the Chan dharma lineage, their family backgrounds, dharma lineages, and life achievements are for the most part as mentioned before. Not counting the thirteen Chan bhikkhuni whose family backgrounds are unclear, about seventy percent of the women Chan monastics listed in dharma lineages were from eminent families in the Jiangnan or southeastern coastal region. Most of these women were also talented poets or writers. More than ninety-five percent were in the dharma lineage of Miyun Yuanwu under his various dharma heirs. This dharma lineage had an abundance of capable persons, and its rich interpersonal and social resources contributed both to the development of Chan and even to the prestige and achievements of individual Chan monastics that allowed for the publication of their recorded sayings and other works and for these works to be included in the Jiaxing Buddhist canon.

While the number of Chan bhikkhuni who were able to be listed in the Chan lineage or have their recorded sayings published in the Jiaxing canon remained far less than that of men, the number of these women who were able to be included in the canon and have their works published, and the number who were able to write poetry and verses, increased sig-

nificantly in the late Ming and early Qing periods. This trend was related to social support systems, such as the increasing literacy rates among women in late imperial China, the flourishing of printing and the publishing industry, and openness of thought. However, in addition to the family backgrounds of these women mentioned before, their personal efforts, the resources they brought together, and the Buddhist support networks of many literati connected to the Chan monastics of their same lineage were also important for their success. In other words, the prestige and success of the men of Chan was not a hindrance but an aid to these women.

I will now use several figures to further illustrate how Chan bhikkhuni were nurtured by a complex interplay of various factors and outline more clearly and comprehensively the close relationship between their achievements and their dharma lineages, family backgrounds, and locations.

First, as mentioned before, I have previously used the *Xu biqiuni zhuan* and the women section of the *Gujin tushu jicheng* to compile a list of sixty-four bhikkhuni who had their works published in the late Ming and early Qing periods. Their birthplaces and areas of activity after becoming bhikkhuni are shown in Figure 7: 'Wanming Qingchu Dongnan Yan-hai 64 Wei Biqiuni Chusheng, Hongfa Huodong de Dili Fenbu Tu' 晚明清初東南沿海64位比丘尼出生、弘法活動的地理分布圖 (The Geographical Distribution of the Birthplaces and Buddhist Teaching of Sixty-Four Bhikkhuni of the Southeast Coast in the Late Ming and Early Qing Period).[14]

The pattern and implications of these data are, as I have explained in a previous article, that it seems that the Jiangnan region was the main area of activity for these bhikkhuni or that they were closely connected to convents of the Jiangnan region. Nearly ninety percent of the literate bhikkhuni were born or were active in cities along the Grand Canal or the coastal cities of Zhejiang or Jiangsu, and a particularly large number came from Suzhou (including Wuxian and Changzhou) and Hangzhou. Clearly, the diversity and openness of these seaport cities provided women with the chance to have their achievements seen and have records left behind about who they were and what they did.

Building on this, I have compiled another figure that depicts the sixty-four Chan bhikkhuni together with thirty-two other bhikkhuni who were recorded in Chan lineages but did not have published recorded sayings. The distribution of their birthplaces and areas of Buddhist teaching is shown in Figure 8: 'Wanming Qingchu nü chanshi (han lieru Chandeng fasi nü chanshi) de shengping zuji' 晚明清初女禪師(含列入禪燈法嗣38位女禪師)的生平足跡 (The Geographical Distribution of Chan Bhikkhuni of the Late Ming and Early Qing Period [Including 38 Female Chan Masters included in Chan Lineage]). The birthplaces and active areas of these women were, with the exception of a few sporadic points along the coast of Huguang, Jiangxi, Fujian, and Guangdong, mostly concentrated in cities along the Grand Canal between Suzhou and Hangzhou, ports, or riverside or waterway-adjacent areas on the southeast coast.

Furthermore, according to Figures 1, 2, 3, 4, 5 and 6c, the distribution of the development of the Linji and Caodong lineages in the late Ming and early Qing periods is shown in Figure 9: 'Mingmo Qingchu Linji yu Caodong famai fenbu' 明末清初臨濟與曹洞宗法脈分佈 (The Distribution of the Lineages of the Linji and Caodong Denominations in the Late Ming and Early Qing Period).[15] This shows clearly the respective developments of the two denominations of Chan Buddhism. The branches of the Caodong denomination were not nearly as extensive as those of the Linji denomination that covered almost all of China. However, in terms of the area of distribution, to say nothing for the moment of the areas outside of China, the dharma lineages of the Linji denomination were, in addition to the concentrated areas of the Jiangnan and southern coastal regions, also active in important ways in Sichuan, Yunnan, Guizhou, Huguang, and northern Henan. The development of the Linji denomination in northern or northeast China seems more sporadic, with the exception of Beijing. The Jiangnan region and coastal areas were also important centers of the Caodong lineage, and it seems to have been particularly thriving compared to the Linji denomination in areas such as Fujian, Guangdong, and central and southern Henan,

where we can see a clear regional contrast and segmentation between the two. There is also some activity of the Caodong denomination in Shanxi and Northeast China.

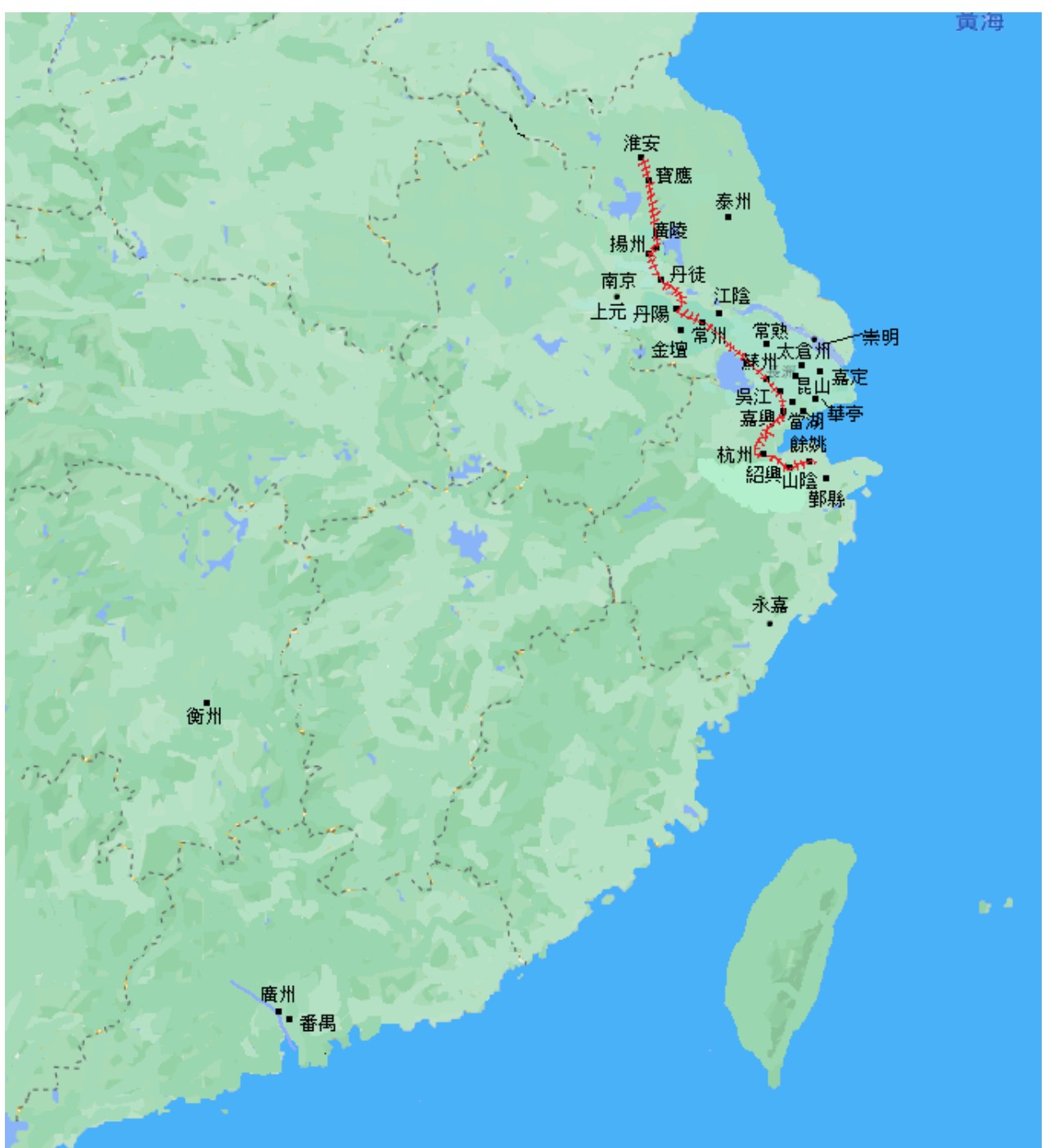

**Figure 7.** The geographical distribution of the birthplaces and Buddhist teaching of sixty-four bhikkhuni of the southeast coast in the late Ming and early Qing periods.

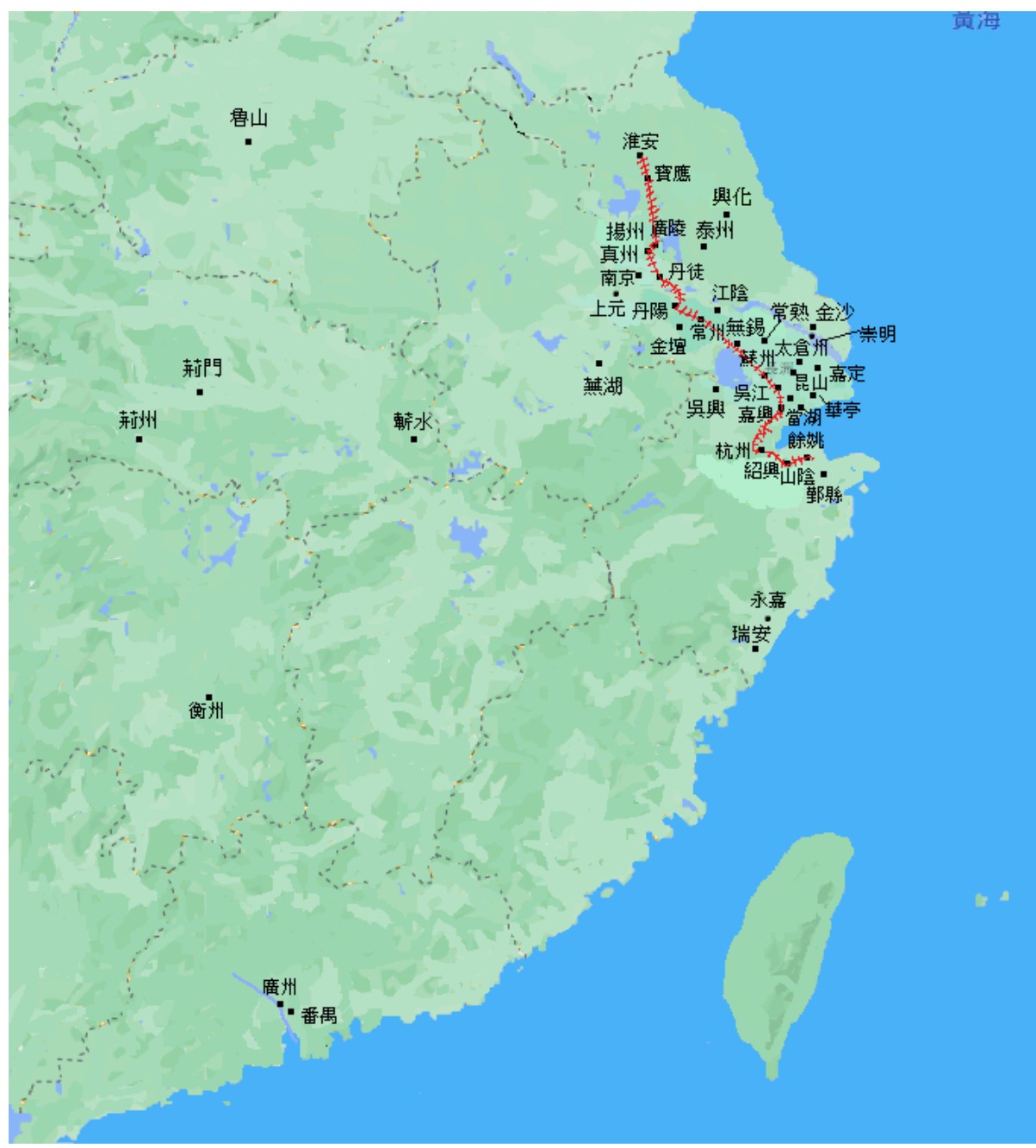

**Figure 8.** The geographical distribution of Chan bhikkhuni of the late Ming and early Qing periods (including 38 female Chan masters included in the Chan lineage).

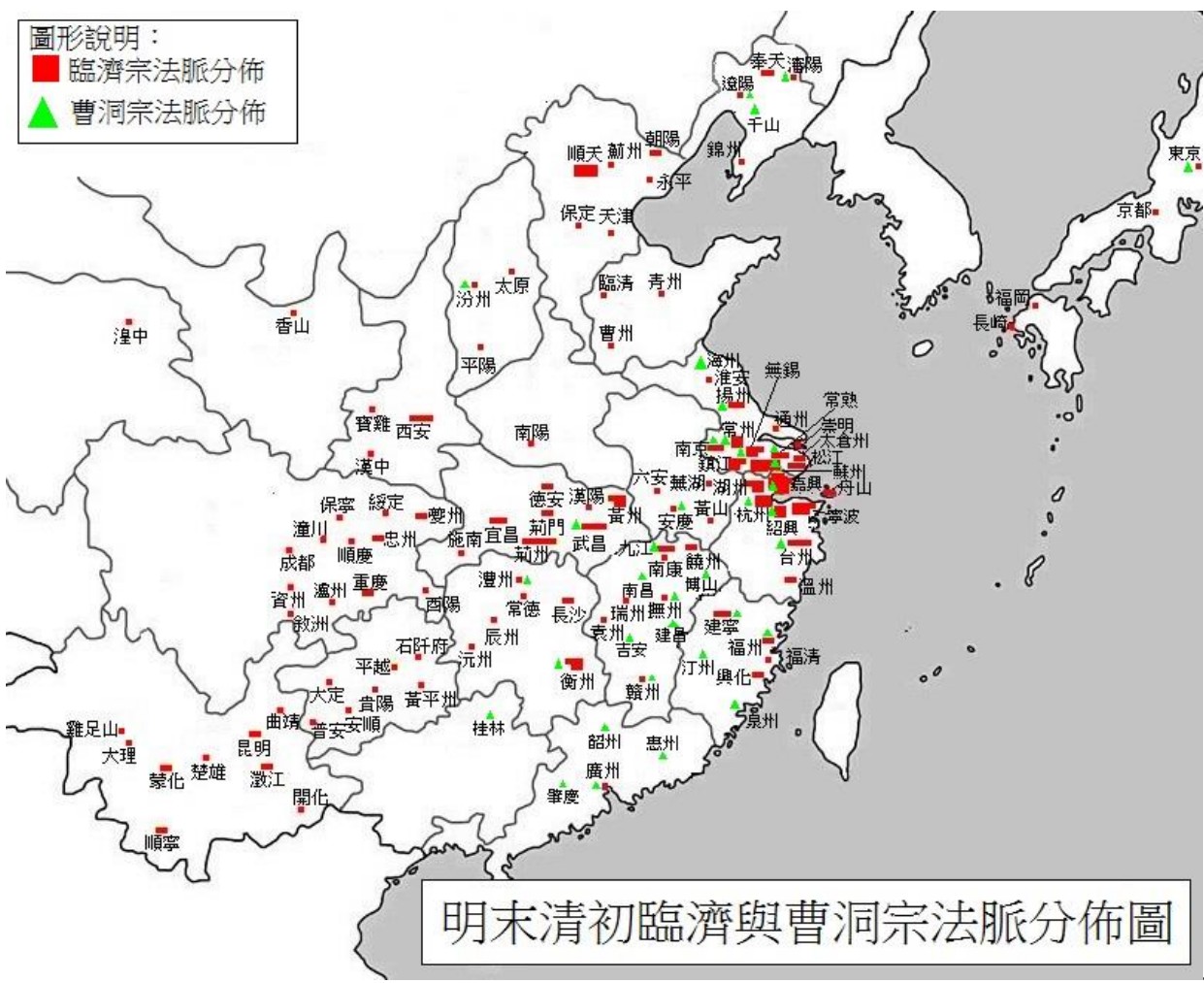

**Figure 9.** The distribution of the lineages of the Linji and Caodong denominations in the late Ming and early Qing periods.

Figure 9 shows that Jiangnan cities along the Grand Canal, ports, rivers, or the coast were important areas where the two denominations overlapped in their development. Such cities include Yangzhou 揚州, Changzhou 常州, Suzhou 蘇州, Jiaxing 嘉興, Hangzhou 杭州, Jiangyin 江陰, Changshu 常熟, Jiading 嘉定, and Yinxian 鄞縣 (Ningbo 寧波). Many of these cities overlap with the active areas of the Chan bhikkhuni of Figures 7 and 8.

For the distribution of the various dharma lineages of the Linji denomination—the lineages of Chuiwan Guangzhen, Yuanyi Miaoyong, Miyun Yuanwu, and Tianyin Yuanxiu—see Figure 10: 'Mingmo Qingchu Linji zong famai fenbu' 明末清初臨濟宗法脈分佈 (The Distribution of the Lineages of the Linji Denomination in the Late Ming and Early Qing Period). Here we find that these dharma lineages continue to be concentrated in Jiangnan cities along the Grand Canal, ports, riverside, or coastal areas, while others are seen in Hubei, Hunan/Hengzhou, northern Henan, Xi'an, Beijing, etc. Sichuan was an important area for the lineages of both Chuiwan and Poshan, which also extended into Yunnan, Guizhou, and northern Henan. The regional development of these lineages seems to be closely related to the birthplace of their Chan monastic members. For example, Fushi Tongxian's lineage is also seen in the Yunnan region. As for the development of each dharma lineage, a general outline can be seen in Figure 10.

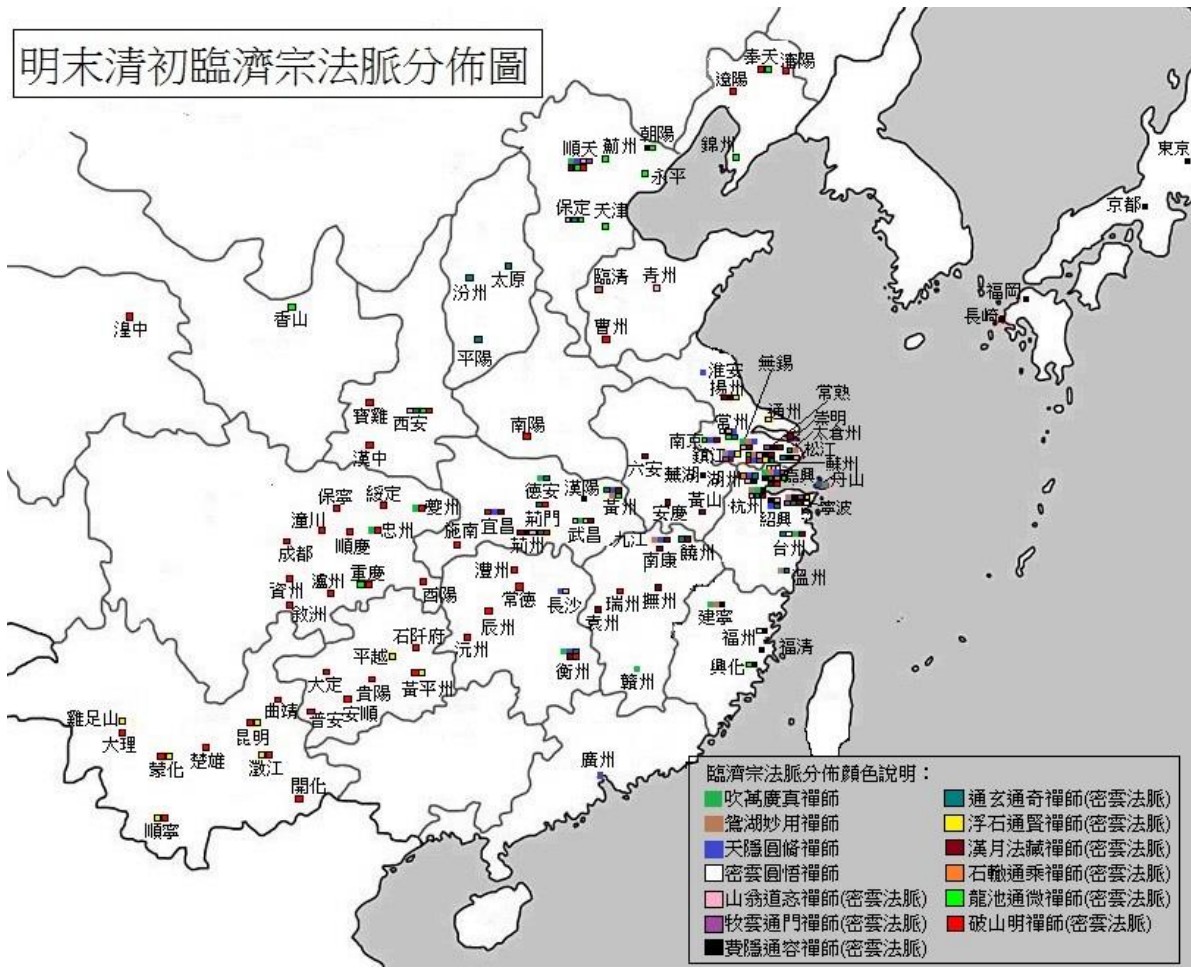

**Figure 10.** The distribution of the lineages of the Linji denomination in the late Ming and early Qing periods.

Whether of the Linji denomination, of the Caodong denomination, of any particular dharma lineage of the Linji denomination, or in regard to the area of activity of a Chan bhikkhuni, the areas along the Grand Canal, ports, riversides, and coastal areas of the Jiangnan region were all major centers for the development of the various Chan monastics. I have used the social resources that went into publishing the Jiaxing Buddhist canon, such as sponsoring monastics and layperson believers who engraved woodblocks, collated, wrote, etc., as well as the publishing sites of the north and south, to show clearly that the most support for the publishing of this canon came from the many cities of the Jiangnan region. See Figure 11: 'Jiaxing zang nanbei zhukechang zhi shehui ziyuan fenbu tu (1589–1644)' 嘉興藏南北諸刻場之社會資源分布圖 (1589–1644) (The Distribution of Social Resources of Sites of Publication of the Jiaxing Buddhist Canon in North and South, 1589–1644).[16] Building on Figure 11, I have compiled a more detailed distribution of cities supporting the publication of the Jiaxing canon (see Figure 12: 'Jiaxingzang Jiangan Kechang, Shikezhe, Kegong, Shuxiezhe, Jiaoduizhe de fenbu' 《嘉興藏》江南刻場、施刻者、刻工、書寫者、校對者的分布 (The Distribution of the Sites of Publication of the Jiaxing Buddhist Canon in the Jiangnan Region: Sponsors, Engravers, Writers, and Collators) (Chen 2022, p. 309).

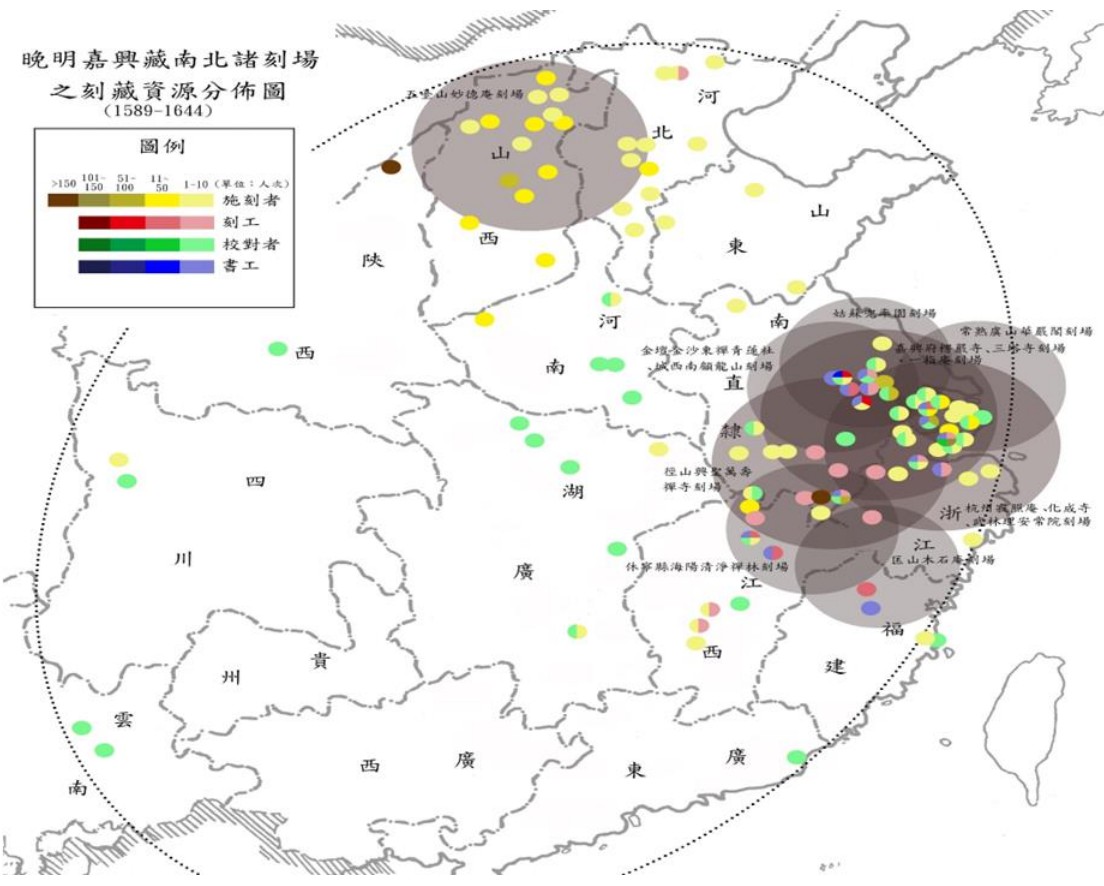

**Figure 11.** The distribution of social resources of sites of publication of the Jiaxing Buddhist canon in the north and south, 1589–1644.

We can see in a comprehensive analysis of these various Figures that the Jiangnan region was the most prosperous region in China in terms of its prosperous and abundant transportation, economic development, cultural publishing, talent of Buddhist monastics and laypersons, foreign and domestic trade, and circulation of information; this is especially so for the highly populated cities of this region close to the stretch of the Grand Canal between Suzhou and Hangzhou, along rivers or lakes, and in coastal areas with dense networks of waterways. For Chan bhikkhuni to access the superior cultural environment and social capital of this region not only nurtured their talent as Buddhist monastics but also must have provided them with considerable support and powerful assistance in displaying their achievements, expanding their influence, and enhancing their visibility.

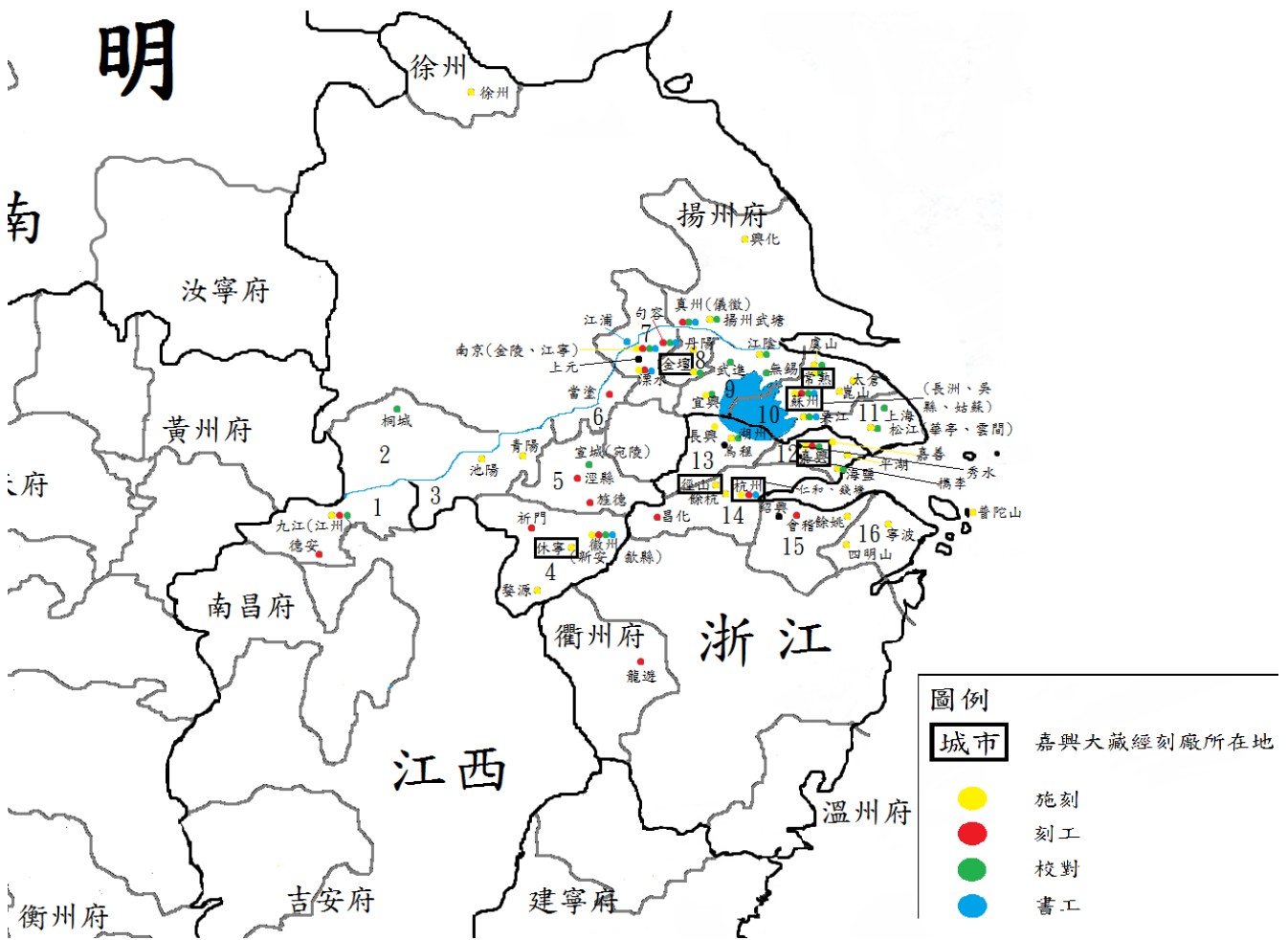

**Figure 12.** The distribution of the sites of publication of the Jiaxing Buddhist canon in the Jiangnan region: sponsors, engravers, writers, and collators.

## 6. Conclusions

As I have already mentioned several times before, this article is a continuation of my previous article 'Wanming Qingchu Dongnan yanhai gangkou Fosi de biqiuni shenying', and it can be regarded as a supplementary work in the same series as that article. Living in open and diverse port cities affected the Buddhist practice and personal attainments of bhikkhuni in different ways. Some of these bhikkhuni tended toward conservatism, strict precept observation, and complete severance of all contacts with the outside world to guard against the tempest of desire and extravagant consumption of the materialistic cities as an interference and encroachment on Buddhism. However, many of these bhikkhuni took advantage of the multiple favorable conditions of urban society to not only achieve their personal goals of Buddhist practice but also spread their influence and reveal their outstanding achievements. In this study, I have thoroughly investigated the lives, achievements, and monastic and layperson relationships of seven Chan bhikkhuni whose works were included in the Jiaxing Buddhist canon, and in doing so, I have shown clearly that the regional, family, and dharma connections were extremely important factors for the success of Chan bhikkhuni.

I have focused in this article on thirty-eight Chan bhikkhuni who were included in the Chan lineages, including the aforementioned seven. All these women were the heirs of various lineages of the Linji denomination of Chan Buddhism. Apart from these seven Chan bhikkhuni for whom we have detailed records of their life deeds and dialogues with monastics and laypersons, most of these women have no surviving works that I could cite,

so I have sought clues about the possible backgrounds of these women indirectly by examining their birth families, relatives, and relationships in their dharma lineage. The results of this investigation are that, not counting a few of these women whom I did not have enough information to identify, most or about seventy percent of these women were born to cultured, literate, wealthy, and eminent families of the Jiangnan region. These women were literate, and many were gifted poets. These women mostly practiced and taught Buddhism close to where they were born after they become bhikkhuni, further confirming that family, regional, and dharma connections were always important supports for a Chan bhikkhuni that also acted as a protective system that would promote their successful Buddhist practice.

As I wrote in my previous work, it seems that most of these Chan bhikkhuni were the boudoir flowers of prestigious families of the Jiangnan region. This is sufficient to show that being born to a prestigious family in an affluent environment was an important resource for success as a bhikkhuni. These women had a far greater chance of success compared to bhikkhuni from impoverished family backgrounds. Bhikkhuni from prestigious family backgrounds also had more opportunities than ordinary people to come in contact with eminent monks who could officiate their monastic conversions through the powerful assistance and recommendations of family members, relatives, and the eminent literati of their localities (Chen 2022, Section 4).

In the vast area of China where the lineages of the Linji denomination were distributed in the late Ming and early Qing periods, there are no records of Chan bhikkhuni in the vast areas of Sichuan, Yunnan, and Guizhou. With the exception of a few scattered bhikkhuni in Jiangxi, Fujian, and Guangdong, all these women were concentrated in the Jiangnan region or various economically prosperous cities of the southeast coast, where there was an abundant and diverse flows of people, goods, and information. There were many eminent and refined literati in the Jiangnan region, and Figures 2 and 3 show that talented Buddhists of various lineages also congregated here. Therefore, most Chan bhikkhuni were born and were active here. The immense concentration of Buddhist resources in the Jiangnan region, whether in terms of personal connections or in terms of material resources, undoubtedly provided more opportunity for Chan bhikkhuni to have a voice. It was especially by having their sayings recorded, compiled, and published that these women were able to enhance their visibility and establish their position in Chan. In other words, the Jiangnan region was a diverse and open regional society full of various possibilities, opportunities, and advantages that fostered the remarkable women Chan monastics who were so rare in other regions.

However, the many talented women of prestigious families who became Buddhist monastics to study Chan in the late Ming and early Qing periods do not seem to have all had the same motive for doing so. Some were driven by their aspirations, family difficulties, or a painful life event. However, the turmoil of the Ming–Qing transition was the most common impetus for both literati men and literati women of that time to leave the world behind for Chan. Most of them made this choice after the conquest of their country and the ruin of their family. Still, when the literati women of prestigious families become bhikkhuni, they certainly must have added the bhikkhuni community in the literacy of Buddhist scripture, the understanding of principles, the propagation of written wisdom, and the promotion of the ability to record dharma talks, such as in the compilations of recorded sayings. They also introduced a new way of engaging in dialogues and expressing thought by using women writing to record the biographical details of the bhikkhuni community and the dharma talks of Chan monastics. When these women wrote Chan poems and verses, they revealed their independent thinking and an enlightened and autonomous side of themselves.

The participation of Confucian literati in Buddhist discourse during the late Ming brought new life to Buddhism by inspiring Buddhists to intellectual pursuits.[17] Similarly, in the late Ming and early Qing periods, when a wave of literate and talented flowers of the boudoir all entered Buddhism at once, there seemingly began a new movement that was different from the staid, hierarchical, gender-specific, and conservative character of the bhikkhuni order established under the traditional patriarchal framework. The new style of the bhikkhuni order that emerged was led by women monastics of the Chan school, and it emphasized equal questioning and discussion in Buddhism by men and women, unrestricted openness, and intellectual independence. Perhaps it can be said that the addition of this group of literati women added a new elegance to Buddhism in the late Ming and early Qing periods.

**Supplementary Materials:** The following supporting information can be downloaded at: https://www.mdpi.com/article/10.3390/rel14101241/s1.

**Funding:** This research was funded by National Science and Technology Council (grant number: 110-2410-H-006-003-MY2).

**Institutional Review Board Statement:** Not applicable.

**Informed Consent Statement:** Not applicable.

**Data Availability Statement:** Not applicable.

**Acknowledgments:** This is a translation of 晚明清初女禪師的出身、法脈及其成就 originally published in Chinese by Cambria Press in the Hualin International Journal of Buddhist Studies 5.2 (2022), pp. 233–292. This translation was prepared by Joseph C. Williams with support from National Science and Technology Council (grant number: 110-2410-H-006-003-MY2). Permission was granted by the Hualin International Journal of Buddhist Studies. License files can be found in the Supplementary Materials.

**Conflicts of Interest:** The author declares no conflict of interest.

## Notes

1  For an overview of previous studies on Ming–Qing bhikkhuni, see Chen (2022).

2  See Chen (1967, pp. 24–35). Chen Yuan 陳垣 (1880–1971), here in the section 'Wudeng quanshu zheng' 五燈全書諍 ('The Dispute over the *Wudeng Quanshu*') of the first fascicle of his book *Cao-Ji zhi zheng* 濟洞之諍 (*Chan Disputes*), mentions that the 1693 *Wudeng quanshu* combined two earlier Chan genealogies—the *Wudeng huiyuan* 五燈會元 (*Compendium of the Five Lamp Transmissions*) and the *Wudeng huiyuan zuanxu* 五燈會元纘續 (*Continued Compendium of the Five Lamp Transmissions*)—and then added supplementary content to include thirty-seven generations of Chan genealogy from Qingyuan Xingsi 青原行思 (671–740) and Nanyue Huairang 南嶽懷讓 (677–744). Chen Yuan praised the *Wudheng quanshu* as being the most comprehensive Chan genealogy, a treasure trove that stood in stark contrast to the petty extremism of Feiyin Tongrong's *Wudeng yantong* 五燈嚴統 (*The Completely Ordered Five Lamp Transmissions of Chan Genealogy*).

3  Translator note: For an English translation of this text, see Grant (2017).

4  For a distribution map of the activity of the bhikkhuni of the Ming–Qing period in the seaport cities of southeast China, see Figure 8 in Chen (2022).

5  Zhang (1974, 282:7258). For Huang Chunyao's experiences during the Ming–Qing transition, see Chen (2018, pp. 315–16).

6  'Fulu: Bao Tizhai xiansheng ba wushi lu yun' 附錄: 鮑體齋先生跋吾師錄云 ('Appendix: Mr. Bao Tizhai's Postscript to My Teacher's Record') in Huang (1676).

7  For details of Qian Qianyi's support of the publication of the Jiaxing Buddhist canon, see Chen (2018, pp. 320–63).

8  See the China Biographical Database Project (Ming Qing Women's Writings), https://authority.dila.edu.tw/person/ (accessed on 16 June 2023)

9  Qiu (1997). For details on Qian Qianyi's friendships with the Xiao Boyu family of Taihe, the Mao Jin family of Changshu, and their sponsorship of the later publication of the Jiaxing Buddhist canon, see Chen (2018, pp. 323–25) and Lin (2018, pp. 45–64).

10  See 'Sun Xiaoruo gao xu' 孫孝若稿序 ('Preface to the Works by Sun Xiaoruo') in Wu (1975, 34:598–99).

11  For the dharma lineage of Fachuang Xingzhi, the dharma heir of Shiqi Tongyun, see Jilun (1975–1989, 75:386).

12  See 'Yinyue', on the China Biographical Database Project website, https://cbdb.fas.harvard.edu/cbdbapi/person.php?name=Yinyue (accessed on 16 June 2023).

[13]   See 'Author [Yinyue]' on the Ming Qing Women's Writings website, Details—Poet :: Ming Qing Women's Writings Digitization Project (mcgill.ca) (accessed on 16 June 2023)

[14]   See Map 8 in Chen (2022).

[15]   The numbers of Linji and Caodong Dharma lineages on which this Figure is based are as follows:

(1)   The monks in the four Linji dharma lineages (of Chuiwan Guangzhen, Yuanhu Miaoyong, Tianyin Yuanxiu, and Miyun Yuanwu) number 430, with the locations of 232 of them identified;

(2)   The monks in the two Caodong dharma lineages (of Zhanran Yuancheng and Wuming Huijing) number 60, with the locations of 44 of them identified'

(3)   A total of 490 Linji and Caodong monks covered, with the locations of 276 of them identified.

[16]   See Map 5-1 in Chen (2010).

[17]   Araki Kengo 荒木見悟 (1917–2017) writes that one of the key factors in the revitalization of Buddhism during the Ming period was the open attitude in Confucianism that came with the rise in Ming Confucianism of Wang Yangming's 王陽明 (1472–1529) philosophy of the heart–mind. This led Confucians to abandon the chauvinistic attitude of Confucian orthodoxy and absorb a great deal of Buddhist and Daoist thought and allowed more space for interactions between Confucians and Buddhists, which stimulated the intellectual advancement of Buddhists. See Araki (1961, p. 1).

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
