# Peer review of "The Social Backgrounds, Dharma Lineages, and Achievements of Women Chan Masters, 1572–1722"

_religions, doi:10.3390/rel14101241_

Round 1
Reviewer 1 Report
This is an original, rich, and well-argued article. I am deeply impressed by the primary materials and the effective methods the author uses throughout it. Nonetheless, there are some minor errors, which the author can find in the PDF file, and I hope that he/she can correct them correspondingly.

Author Response
I really appreciate the reviewer for his/her careful reading of my article and for giving me practical suggestions to improve it. I found that most of the errors are minor and associated with the translation, and have thus corrected all of them according to the advice.
Reviewer 2 Report
This is a very well researched and written paper. Thank you! I have noticed a few minor issues that you may wish to correct prior to publication. They include the following:
Line 145
Correct “their” to “there”
Line 1103
The expression “boudoir flower” seems unusual in English. Is this a translation of a Chinese expression? If so, you might add a gloss explaining this.
Line 1120
Perhaps you should change “seven sui old” to “seven years old”?
Line 1247
Correct “znag” to “zang”
It is excellent; I only found a few minor issues.
Author Response
My thanks are to the reviewer for his/her careful reading and advice. Below is my response to his/her suggestions to improve the article:
1、Line 145 Correct “their” to “there”
Response: It is surely difficult to distinguish "of their being" from "of there being", but after checking the context carefully, I would like to keep the sentence here unchanged and leave it to the judgment of the editor.
2、Line 1103 The expression “boudoir flower” seems unusual in English. Is this a translation of a Chinese expression? If so, you might add a gloss explaining this.
Response: The term boudoir flower here is not a translation of a Chinese expression. Basically, it means an outstanding girl. So, I don't think it is necessary to add a gloss here.
3. Line 1120 Perhaps you should change “seven sui old” to “seven years old”?
Response: It is confusing to decide to use "sui" or "years old", but I would like to follow the advice and change the sentence accordingly.
4、Line 1247 Correct “znag” to “zang”
Response: Thanks for the reminder. I have changed the word.